# Glycoside hydrolases enhance antibiotic activity against *Pseudomonas aeruginosa* biofilms from cystic fibrosis airways

Isaac Martin,[1,2] Jonathan Chung,[2,3] Deepa Raju,[4] Yvonne Yau,[2] Amanda J. Morris,[2,5] Shafinaz Eisha,[2] P. Lynne Howell,[4,6] Valerie Waters[2,7]

**ABSTRACT** *Pseudomonas aeruginosa* forms antibiotic-tolerant biofilms in cystic fibrosis (CF) lungs. Targeting exopolysaccharides such as Psl and Pel offers a potential strategy to disrupt biofilms and improve antibiotic efficacy. Fourteen clinical CF *P. aeruginosa* isolates were screened for Psl- and Pel-dependent biofilm formation using crystal violet assays. The adjunctive effects of glycoside hydrolases (GHs) $PslG_h$ and $PelA_h$ with colistin, levofloxacin, and tobramycin (at peak sputum concentrations following nebulization) were evaluated using a high-throughput assay. Confocal microscopy visualized effects in live biofilms at subinhibitory antibiotic levels for two representative strains. Enzyme stability in CF sputum supernatant at 37°C was assessed via Western blot. Nine of 14 isolates showed $PslG_h$-mediated disruption; only 3/14 were disrupted by $PelA_h$. $PslG_h$ significantly enhanced antibiotic effects in 24 of 42 strain-antibiotic pairs (57%), compared with 5 of 42 (12%) for $PelA_h$. Adjunctive effects were most pronounced with tobramycin, yielding 18.1% biofilm reduction. Confocal imaging confirmed additive effects when the GH corresponded to the exopolysaccharide to which the isolate had demonstrated disruption in the screening assay, but showed no benefit when targeting the other exopolysaccharide, even at higher enzyme doses. $PslG_h$ was more stable than $PelA_h$ in sputum, with $PelA_h$ degraded significantly by 4 and 20 h. GHs targeting exopolysaccharides associated with biofilm disruption in functional assays can enhance antibiotic efficacy against *P. aeruginosa* biofilms. $PslG_h$ shows greater promise due to broader activity and better sputum stability. These findings support tailored GH-antibiotic strategies in CF lung infection treatment.

**IMPORTANCE** Biofilms formed by *Pseudomonas aeruginosa* are a major obstacle to effective antibiotic treatment, particularly in chronic lung infections encountered in cystic fibrosis. This study explores the use of glycoside hydrolases (enzymes that degrade key biofilm components) as a way to enhance antibiotic activity against early-stage clinical isolates. Our findings show that targeting biofilm structure can potentiate the effects of multiple antibiotics, suggesting that biofilm-degrading compounds hold promise as future antibiotic adjuvants. However, variability among bacterial strains highlights the need for further research before these strategies can be reliably translated into clinical care.

**KEYWORDS** glycoside hydrolase, antibiotic adjuvant, biofilm

*P*seudomonas aeruginosa is an ubiquitous, Gram-negative opportunistic pathogen that infects vulnerable patient populations (1). Individuals with the autosomal recessive multisystem disease cystic fibrosis (CF) are at risk of significant *P. aeruginosa* lung infection due to dehydrated and viscous mucus in the airways (2). Although antibiotic use has contributed to improved outcomes (3), *P. aeruginosa* infection remains a major contributor to CF-related mortality (4, 5). Over time, *P. aeruginosa* adapts

**Peer Reviewer** Michael D. Parkins, University of Calgary, Calgary, Alberta, Canada

Address correspondence to Isaac Martin, isaac.martin@sickkids.ca.

This work was supported in part by grants from GlycoNET no. ID-03 and ID-11 to V.W. and P.L.H., and the Canadian Institutes of Health (CIHR) no. FDN154327 to P.L.H. P.L.H. was the recipient of a Tier 1 Canada Research Chair (2006-2020).

See the funding table on p. 14.

genetically and phenotypically to the CF lung environment in response to selective pressures, such as inflammation and antibiotic exposure. These adaptations promote multidrug resistance (MDR), biofilm formation, and a chronic, host-adapted phenotype, complicating treatment and worsening prognosis (6, 7).

In the airways, *P. aeruginosa* forms biofilm aggregates suspended in sputum, embedded within a matrix of extracellular polymeric substances (EPS), including exopolysaccharides, proteins, and extracellular DNA (eDNA) (8, 9). Although inhaled antibiotics can achieve airway concentrations far exceeding those of systemic delivery (10, 11), the EPS matrix provides physical protection against antimicrobials with biofilm-embedded *P. aeruginosa* exhibiting minimum inhibitory concentrations (MICs) that are significantly higher than those of their planktonic counterparts (12, 13). Exopolysaccharides support cell adhesion and aggregation, enhancing bacterial survival under antibiotic pressure (14, 15). Several exopolysaccharides contribute to this protective matrix, including alginate, Psl, and Pel. Alginate is a hallmark of mucoid strains and is strongly associated with chronic CF infections, contributing to immune evasion and biofilm persistence (16). Disrupting these structural elements to improve antibiotic penetration is a promising adjuvant strategy.

One such approach employs glycoside hydrolases (GHs) that target the matrix exopolysaccharides Psl and Pel, which are detected in CF sputum biofilms of *P. aeruginosa* (17). Psl, a neutral sugar polymer, facilitates early biofilm development and confers tolerance to colistin, tobramycin, and ciprofloxacin *in vitro* (18–20). Pel, a positively charged polymer, mediates cell-cell interactions and also enhances antibiotic tolerance (20–22), likely through its association with negatively charged eDNA, which stabilizes the matrix and limits drug penetration (17). Biofilms are dynamic, and exopolysaccharide biosynthesis operons typically encode specific, substrate-adapted GHs to regulate biofilm remodeling (23). Exogenous administration of such enzymes—Psl glycoside hydrolase ($PslG_h$) and PelA hydrolase ($PelA_h$)—has shown potent biofilm-disruptive effects with nanomolar activity *in vitro* and *in vivo* (24, 25).

Previous studies have demonstrated that $PslG_h$ and $PelA_h$ potentiate antibiotics across models. In a wound infection model, their combination enhanced the activity of ciprofloxacin, tobramycin, colistin, neomycin, and polymyxin B, while also improving immune activity with increased complement deposition (26). In murine infection models, co-administration of $PslG_h$ and $PelA_h$ improved ciprofloxacin efficacy but not ceftazidime (27), while $PelA_h$ in a cellulose dressing disrupted wound biofilms (28). Despite these promising results, the relative importance of Psl and Pel across diverse clinical isolates is unclear. Many studies have used laboratory strains or small isolate panels, and antibiotic exposures do not always reflect CF airway conditions. Moreover, the stability of these enzymes in CF sputum remains poorly characterized. These factors could all have downstream implications for clinical utility.

In this study, we used functional disruption assays to compare the relative functional dependence of early *P. aeruginosa* CF isolates on Psl versus Pel during biofilm formation, and to evaluate whether enzymatic disruption of these matrix components could enhance antibiotic activity. We examined a range of *P. aeruginosa* CF isolates and found that $PslG_h$ consistently reduced biofilm biomass more than $PelA_h$, indicating greater Psl reliance. There were instances where $PslG_h$ and $PelA_h$ potentiated colistin, levofloxacin, and tobramycin across genetically distinct strains at concentrations achievable via nebulization. Confocal microscopy imaging with live-cell biomass quantification confirmed these effects in representative Psl- or Pel-reliant strains. Lastly, we found that $PslG_h$, but not $PelA_h$, remained stable in CF sputum supernatant at 37 °C.

## MATERIALS AND METHODS

### Selection of *Pseudomonas aeruginosa* isolates, storage, and reanimation

Fourteen first-time *P. aeruginosa* clinical isolates from distinct pediatric CF patients at the Hospital for Sick Children were selected to provide a broad representation of early-infection phenotypes, extending beyond the limited number of clinical strains previously tested in GH–antibiofilm studies (21, 23, 25–30). Seven were from patients who subsequently developed chronic infection despite standard eradication therapy, and seven were from patients in whom infection was successfully eradicated, as previously described (31, 32). Isolates were retrieved from −80°C storage and streaked on blood agar (Oxoid, Nepean, ON, Canada), then incubated overnight at 37°C. Following three subcultures as per standard microbiological practice (33), a single colony was inoculated into 4 mL lysogeny broth (LB; BioShop, Burlington, Canada) and incubated overnight at 37°C with shaking (225 rpm). A 1:100 dilution of this culture was transferred into fresh LB and grown to early logarithmic phase, approximately 0.1 optical density (OD) at 600 nm. These suspensions were used for 96-well plate assays and chamber slide experiments.

### Use of PslG and PelA glycoside hydrolases

The recombinant GHs $PslG_h$ and $PelA_h$ were purified as described previously via nickel-affinity chromatography, followed by buffer exchange into 20 mM Tris (pH 7.5), 150 mM NaCl, and 2% (vol/vol) glycerol for $PslG_h$, and 20 mM Tris (pH 8.0), 150 mM NaCl, and 10% (vol/vol) glycerol for $PelA_h$ (25, 29). The GHs were stored at −80°C and diluted in 1× PBS to final working concentrations of 0.1 mg/mL (crystal violet assays) and 0.1 or 0.5 mg/mL (confocal experiments). This is consistent with concentrations used in previous studies disrupting biofilms to enhance antibiotic efficacy *in vitro* and *in vivo* (25–27).

### Choice of antibiotics and concentrations

Three inhaled antibiotics commonly used in CF care were tested: colistin (colistin sulfate; Sigma-Aldrich, Oakville, ON), levofloxacin (Santa Cruz Biotechnology, Mississauga, ON), and tobramycin (Thermo Fisher Scientific, Ottawa, ON). Stock solutions were prepared following the manufacturer's instructions. Tobramycin sulfate was dissolved in sterile water at 50 mg/mL, filter-sterilized (0.22 µm), aliquoted, and stored at −20°C. Colistin sulfate was dissolved in sterile water at 10 mg/mL and prepared fresh weekly. Levofloxacin was prepared at 50 mg/mL in sterile water with mild NaOH adjustment to facilitate dissolution (final pH 6.8–7.2), then filter-sterilized and stored at −20°C protected from light. Initial working concentrations used in assays were chosen to reflect upper-airway levels reported after nebulization: colistin, 40 µg/mL; levofloxacin, 4,500 µg/mL; and tobramycin, 1,000 µg/mL (34–36).

### Screening strains for biofilm disruption by $PslG_h$ or $PelA_h$

To determine relative contribution of Psl or Pel to biofilm formation, isolate cultures in early log phase were pipetted (200 µL) into 96-well plates (Thermo Fisher Scientific, Mississauga, ON) and incubated statically at 37°C for 24 h. After removing planktonic cells, 180 µL of LB ± antibiotic and 20 µL of $PslG_h$ or $PelA_h$ (final concentration 0.1 mg/mL) were added. Plates were incubated at 37°C for an additional 18 h. Wells were rinsed twice with tap water, tapped dry, and then stained with 180 µL of 0.5% crystal violet (Sigma-Aldrich, Oakville, ON) for 15 min at room temperature. Plates were rinsed three times, dried, and 200 µL of 95% ethanol was added to solubilize biofilm. After 30 min, absorbance at 550 nm was measured using a microplate reader (Varioskan LUX, Thermo Fisher Scientific, Mississauga, ON). This approach was chosen to allow high-throughput screening across multiple clinical isolates. Direct chemical analysis of matrix polysaccharides was not performed due to its complexity and limited feasibility in translational or clinical applications.

## *Pseudomonas aeruginosa* biofilm growth in glass slide chamber

Two clinical isolates were selected for confocal analysis based on Psl (PA342) or Pel (PA380) GH disruption. Early log-phase cultures (220 µL) were added to wells of Nunc Lab-Tek II chambered coverglass slides (Thermo Fisher, Mississauga, ON) and incubated statically for 24 h at 37°C. Media were removed and replaced with 220 µL LB (control) or 180–200 µL LB with 20 µL of antibiotic ± 20 µL of GH (final well volume: 220 µL).

Two subinhibitory antibiotic concentrations were selected for each agent. To assess the effects of GH-antibiotic combinations on biofilm architecture and cell viability, we selected strain-specific "high" and "low" antibiotic concentrations for confocal microscopy experiments. For each isolate, we began by testing antibiotics at the peak sputum concentrations used in the previous experiments. If no viable biofilm structure or live cells were observed by confocal microscopy with live-cell staining at this concentration, we performed serial two-fold dilutions until a viable biofilm was detectable. The highest concentration at which a viable biofilm could still be observed was designated the "high" concentration. The "low" concentration was defined as 20% of this high value. For isolate PA342, the low and high antibiotic concentrations (µg/mL) were colistin (8, 40), levofloxacin (100, 500), and tobramycin (100, 500). For isolate PA380, the corresponding concentrations were colistin (8, 40), levofloxacin (100, 500), and tobramycin (200, 1000). In cases where biofilm viability was still present at the initial peak sputum concentration, this value was taken as the "high" concentration, and the "low" was set at 20% of that value. Hydrolases were also tested at low and high concentrations of 0.1 and 0.5 mg/mL, respectively, to evaluate whether higher enzyme levels produced greater biofilm disruption. In parallel, combination GH treatment (PslG$_h$ + PelA$_h$ at 0.1 mg/mL or 0.5 mg/mL each) was also tested in both strains to assess potential additive effects.

## Confocal imaging and quantification of live-cell biovolume

After 18 h of incubation at 37°C, wells were washed gently, and biofilms were stained with the LIVE/DEAD BacLight bacterial viability kit (SYTO 9 and propidium iodide; Thermo Fisher, Mississauga, ON) in LB for 45 min at room temperature. Wells were then gently washed and replaced with 200 µL of fresh LB for imaging. LB was selected over buffer-based media to avoid biofilm collapse or matrix rearrangement associated with nutrient withdrawal prior to imaging. Both channels were acquired using identical laser and gain settings on a Quorum confocal laser scanning microscope (Zeiss AxioVert 200 M, 25× objective; total magnification ×250; 0.3 µm z-steps). Excitation was set at 488 nm (SYTO 9) and 561 nm (propidium iodide), with emission collected at 500–540 nm and 600–650 nm, respectively. Laser power and detector gain settings were kept constant across all experimental conditions. The PI channel was used during acquisition to visually identify and adjust for PI-positive (non-viable) material, and quantification was then performed from the SYTO 9 channel under fixed imaging parameters across all conditions. Image stacks were processed in Velocity software.

The integrated SYTO 9-positive voxel volume (µm$^3$) was recorded as "live-cell biovolume," reflecting the biomass of membrane-intact cells visualized in the green channel. This metric was supported by corresponding CFU assays performed on parallel biofilms grown on glass chamber slides under identical conditions. Following treatment, biofilms were scraped into PBS, serially diluted, and plated on LB agar for colony enumeration using standard methods (37). While complete recovery of all biofilm-associated cells cannot be assumed, the same procedure was applied uniformly across all conditions, allowing valid relative comparisons of viable bacterial burden.

## Determining the protein integrity of PslG$_h$ and PelA$_h$ when incubated with CF sputum supernatant

Sputum from CF patients (REB# 1000081709) was collected on ice within 1 h of expectoration and processed to obtain supernatant as previously described (38). Recombinant PslG$_h$ and PelA$_h$ at a concentration of 0.1 mg/mL were incubated in PBS, or

in PBS with 10% or 20% sputum supernatant, at 37°C. Aliquots were collected at 0, 4, and 20 h.

Samples were mixed with loading buffer, boiled, and resolved by SDS-PAGE, followed by Western blotting using rabbit polyclonal antibodies against $PslG_h$ and $PelA_h$ (Cedarlane, Burlington, ON) (27, 30). Goat anti-rabbit horseradish peroxidase-conjugated secondary antibody (Bio-Rad, Mississauga, ON) was added to the blot, and binding was detected with a chemiluminescent substrate (Thermo Fisher Scientific, Ottawa, ON). Detection was performed using goat anti-rabbit HRP-conjugated secondary antibody (Bio-Rad, Mississauga, ON) and chemiluminescent substrate (Thermo Fisher Scientific, Ottawa, ON). Protein stability was quantified via densitometry using ImageJ. Band intensities were normalized to the 0 h time point.

To confirm the enzymatic activity of the GHs under the same conditions used for the Western blots, we performed a functional disruption assay using two laboratory strains with well-defined matrix architectures (PAO1 for $PslG_h$ and PA14 for $PelA_h$). Each GH was pre-incubated with or without sputum supernatant following the same protocol as above, then applied to 24-hour biofilms for 1 h before crystal violet quantification.

## Statistical analyses

All statistical analyses were performed using GraphPad Prism (version 10; GraphPad Software, San Diego, CA). For high-throughput biofilm biomass assays, differences between antibiotic alone and GH-antibiotic combinations were assessed using paired *t*-tests. Because each test involved only two groups, no correction for multiple comparisons was applied. For confocal microscopy experiments, one-way analysis of variance (ANOVA) followed by Dunnett's post hoc test was used to compare each treatment group to the antibiotic-alone control. Data are presented as means ± standard error of the mean (SEM), and $P < 0.05$ was considered statistically significant.

## RESULTS

The 14 clinical *P. aeruginosa* isolates used in this study were from different pediatric patients at the Hospital for Sick Children (Toronto, Canada), and the clinical and demographic characteristics of these patients are detailed in Table S1 at https://zenodo.org/records/18701433. Minimum inhibitory concentration (MIC) values for each isolate to the three antibiotics tested are listed in Table S2 at https://zenodo.org/records/18701433.

## More clinical isolates are reliant on Psl than Pel

We screened 14 early *P. aeruginosa* clinical isolates from individuals with CF using functional enzymatic disruption assays to infer strain-specific dependence on the Psl and Pel exopolysaccharide in mature biofilms. As $PslG_h$ and $PelA_h$ degrade their target biofilm polysaccharides rapidly—within minutes *in vitro* (25, 27)—we evaluated their effect after 1-hour exposure using a crystal violet biofilm assay to quantify changes in biofilm biomass (Fig. 1).

To facilitate comparison across strains, biofilm biomass reduction is presented as a percentage relative to the untreated control for each isolate. However, as baseline biofilm formation varied between isolates, the average $OD_{600}$ values for all 14 strains are provided in Fig. S1 at https://zenodo.org/records/18701433 along with results of the statistical tests performed. Notably, although some replicates did cluster around zero, there was no significant increase in biofilm formation with application of any enzymatic treatment. Application of $PslG_h$ led to significant biofilm reduction in 9 of 14 strains (64%), whereas $PelA_h$ led to significant reduction in only 3 of 14 strains (21%) (Fig. 1a). Overall, mean biomass reduction across all strains was 38.6% (95% CI ± 12.1) for $PslG_h$ versus 9.9% (95% CI ± 5.8) for $PelA_h$ (Fig. 1b). Thus, more tested clinical isolates exhibited $PslG_h$-mediated biofilm disruption. Notably, the strain that responded most strongly to $PelA_h$ (PA380) was one of the few for which $PslG_h$ had little effect, suggesting that Pel may still be a clinically relevant EPS.

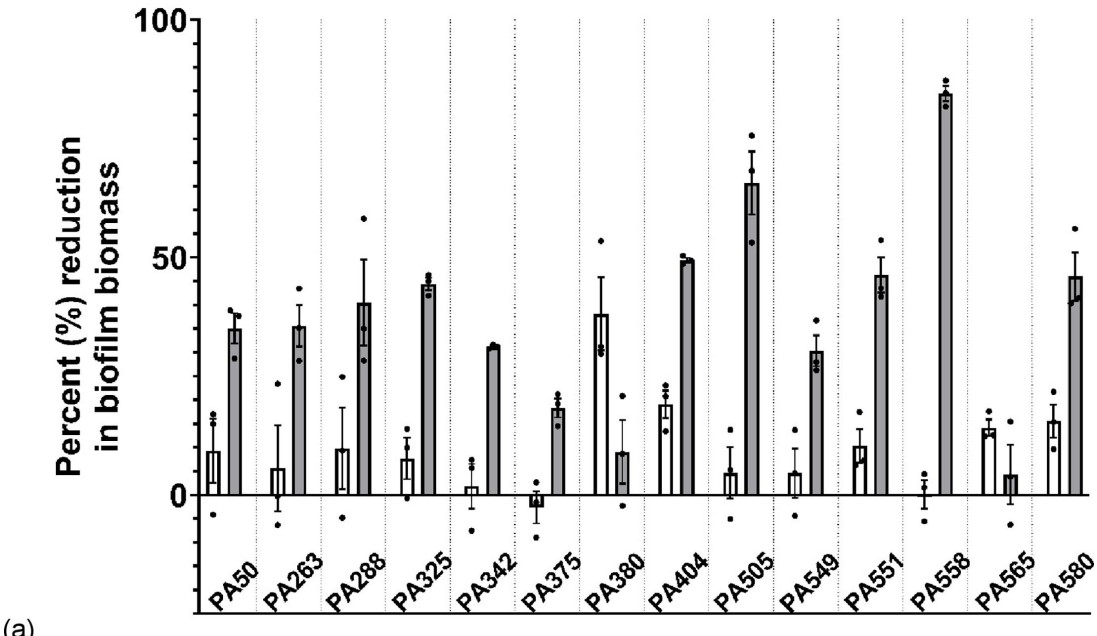

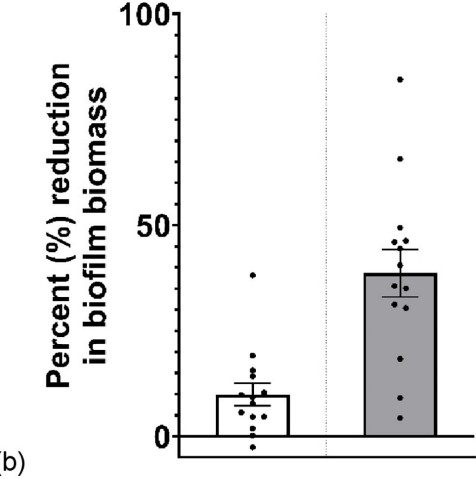

**FIG 1** Biofilm biomass quantification by crystal violet assay across 14 clinical *P. aeruginosa* strains treated with 0.1 mg/mL of PelA$_h$ (white bars) or PslG$_h$ (gray bars). (a) Percent biofilm reduction relative to untreated controls for each strain. Circles indicate means of four technical replicates, with bars representing means ± SEM of three biological replicates. (b) Summary of biofilm reduction across all strains. Circles show means of three biological replicates, with bars representing means ± SEM.

## PslG$_h$ potentiates the effects of inhaled antibiotics more frequently than PelA$_h$

To assess the potential of PslG$_h$ and PelA$_h$ as adjuncts to antibiotics commonly administered via nebulization, we treated pre-formed biofilms of all 14 *P. aeruginosa* clinical isolates with colistin (40 µg/mL), levofloxacin (4,500 µg/mL), or tobramycin (1,000 µg/mL) in combination with each GH (0.1 mg/mL). Biofilm biomass was quantified by crystal violet assay after 18-hour antibiotic/enzyme co-incubation (Fig. 2).

To quantify the incremental effect of glycoside hydrolase co-administration, biofilm biomass reduction was calculated relative to antibiotic-alone controls under matched conditions. Using this approach, PslG$_h$ co-administration resulted in an additional 7.6 ± 10.0% (95% CI), 15.7 ± 9.2% (95% CI; *P* < 0.05), and an 18.1 ± 5.7% (95% CI; *P* < 0.001) reduction in biofilm biomass when combined with colistin, levofloxacin, and tobramycin,

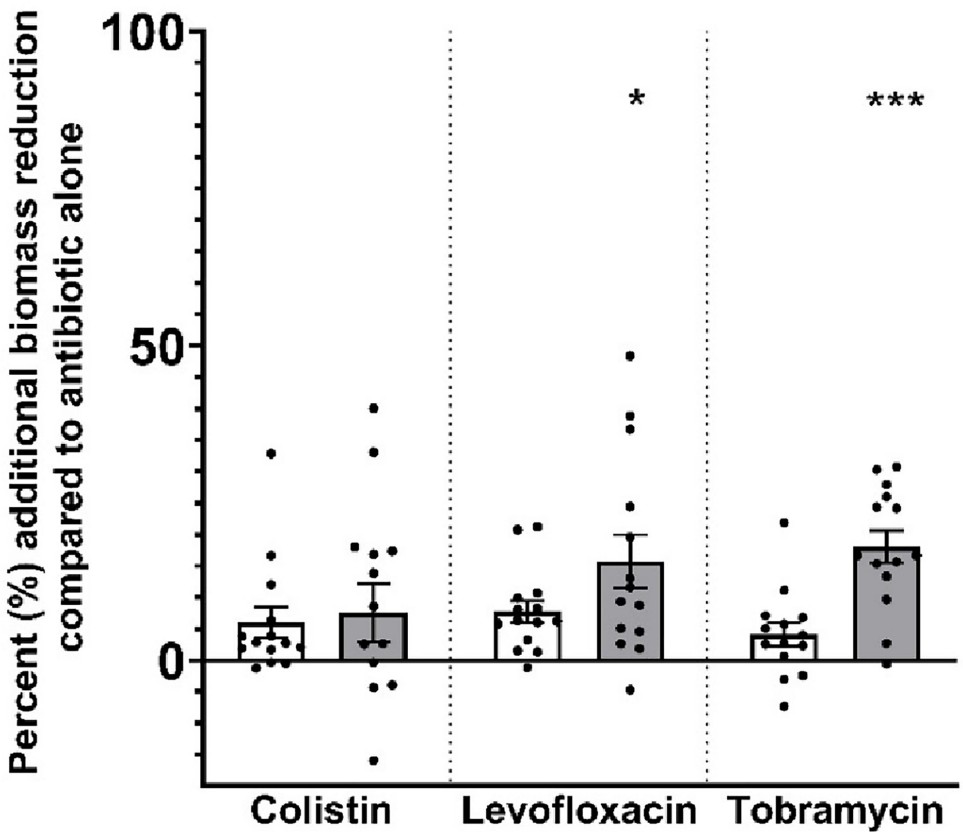

**FIG 2** The adjunctive benefit of co-administration of antibiotics with PelA$_h$ (white bars) and PslG$_h$ (gray bars), assessed by the crystal violet assay when compared to antibiotic alone. The Y-axis shows additional percent biofilm reduction versus antibiotic alone. Bars represent mean ± SEM; each point is the mean of three biological replicates (four technical replicates each). Paired *t*-tests were used to compare antibiotic + enzyme vs antibiotic alone. *$P < 0.05$; ***$P < 0.001$.

respectively. PelA$_h$ co-administration produced additional biofilm reductions of 6.0 ± 5.3% (95% CI), 7.7 ± 3.8% (95% CI), and 4.1 ± 4.0% (95% CI) when combined with colistin, levofloxacin, and tobramycin, respectively.

### Adjunctive effects by strain: PslG$_h$ is more frequently effective than PelA$_h$

We next used paired *t*-tests to compare the effect of antibiotic + GH combination with antibiotic alone for each individual strain (Table 1). We deemed the combination to be additive (A) if the results showed significantly more biofilm biomass reduction, and to have no effect (N) if there was no significant difference. Notably, we did not find any cases of an inhibitory interaction (I), where there was statistically more biofilm adherence when antibiotic and enzyme were used relative to the biomass present with the antibiotic alone. For colistin, 6 of the 14 clinical strains showed significantly more biofilm reduction when PslG$_h$ was added, compared to no strains when the PelA$_h$ was added. For levofloxacin, 6 of the 14 strains showed significantly more biofilm reduction when PslG$_h$ was added, compared to only 3 of 14 strains with the PelA$_h$. For tobramycin, 12 of the 14 strains showed significantly more biofilm reduction with the PslG$_h$, compared to only 2 of 14 strains with the PelA$_h$. Overall, PslG$_h$ yielded significant additive effects in 24 of 42 possible strain-antibiotic combinations (57%), compared to just 5 of 42 (12%) for PelA$_h$. Notably, in 3 of the 5 PelA$_h$-responsive cases, PslG$_h$ was ineffective—highlighting PelA$_h$'s utility in a subset of strains, particularly those with PelA$_h$-disrupted matrices.

TABLE 1 Summary data from biomass quantification assay by strain[a,b]

| Antibiotic | Colistin | | Levofloxacin | | Tobramycin | |
|---|---|---|---|---|---|---|
| Strain | +PslG$_h$ | +PelA$_h$ | +PslG$_h$ | +PelA$_h$ | +PslG$_h$ | +PelA$_h$ |
| Pa50 | N | N | A | N | A | N |
| Pa263 | A | N | A | A | A | N |
| Pa288 | A | N | A | N | A | N |
| Pa325 | N | N | A | N | A | A |
| Pa342 | A | N | N | N | A | N |
| Pa375 | A | N | A | N | A | N |
| Pa380 | N | N | N | A | N | A |
| Pa404 | N | N | A | N | A | N |
| Pa505 | N | N | N | N | A | N |
| Pa549 | N | N | N | N | A | N |
| Pa551 | N | N | N | N | A | N |
| Pa558 | A | N | N | A | A | N |
| Pa565 | N | N | N | N | A | N |
| Pa580 | A | N | N | N | N | N |

[a]N, No effect; A, Additive; I, Inhibitory.
[b]Clear background, No effect; Gray background, Additive effect; Black background, Inhibitory effect.

## Confocal microscopy confirms glycoside hydrolase-antibiotic additivity depends on matrix target specificity

To validate our high-throughput biomass findings and gain more detailed insight into bacterial viability, we employed confocal microscopy with live-cell staining. This approach allowed us to directly visualize live biofilm-associated cells following treatment, in contrast to crystal violet assays which measure total biomass, including dead cells and matrix material. We also tested two antibiotic concentrations to reflect the variable drug levels that may occur in different lung regions following inhalation therapy. For PA342, low/high antibiotic concentrations (µg/mL) were colistin 8/40, levofloxacin 100/500, and tobramycin 100/500; for PA380, concentrations were colistin 8/40, levofloxacin 100/500, and tobramycin 200/1000. Two representative clinical strains were selected based on their susceptibility to GH disruption—PA342 (PslG$_h$-disrupted) and PA380 (PelA$_h$-disrupted)—to assess whether GH activity depends on targeting the exopolysaccharide to which each strain showed the greatest susceptibility in the initial screen, and whether higher enzyme concentrations (0.5 mg/mL vs 0.1 mg/mL) produce additive effects not observed in the initial single-dose screen.

We also tested dual PslG$_h$ + PelA$_h$ treatments on both representative isolates (PA342 and PA380). As shown in Fig. S2 and S3 at https://zenodo.org/records/18701433, combined GH treatment did not result in additional biofilm reduction compared to the single GH alone at equivalent concentrations.

## PslG$_h$-disrupted strain PA342 responds to PslG$_h$ but not PelA$_h$

PA342 was treated with the antibiotics colistin, levofloxacin, and tobramycin at subinhibitory concentrations, and with PslG$_h$ or PelA$_h$ at 0.1 or 0.5 mg/mL using confocal microscopy (Fig. 3).

Significant reductions in live-cell biovolume were observed for all three antibiotics when co-administered with PslG$_h$ under specific concentration conditions (Fig. 3a). In contrast, PelA$_h$ had no significant effect under the same conditions (Fig. 3b). CFU analysis showed reductions in viable cell counts seen with levofloxacin (500 µg/mL) when co-administered with both the low and high PslG$_h$ concentrations (Fig. 3c). Notably, no viable cells were detected when high-dose levofloxacin was combined with high-dose PslG$_h$ (Fig. 3c). Neither live-cell biovolume quantification by confocal microscopy nor CFU analysis demonstrated a consistent dose-dependent effect of the relevant GH.

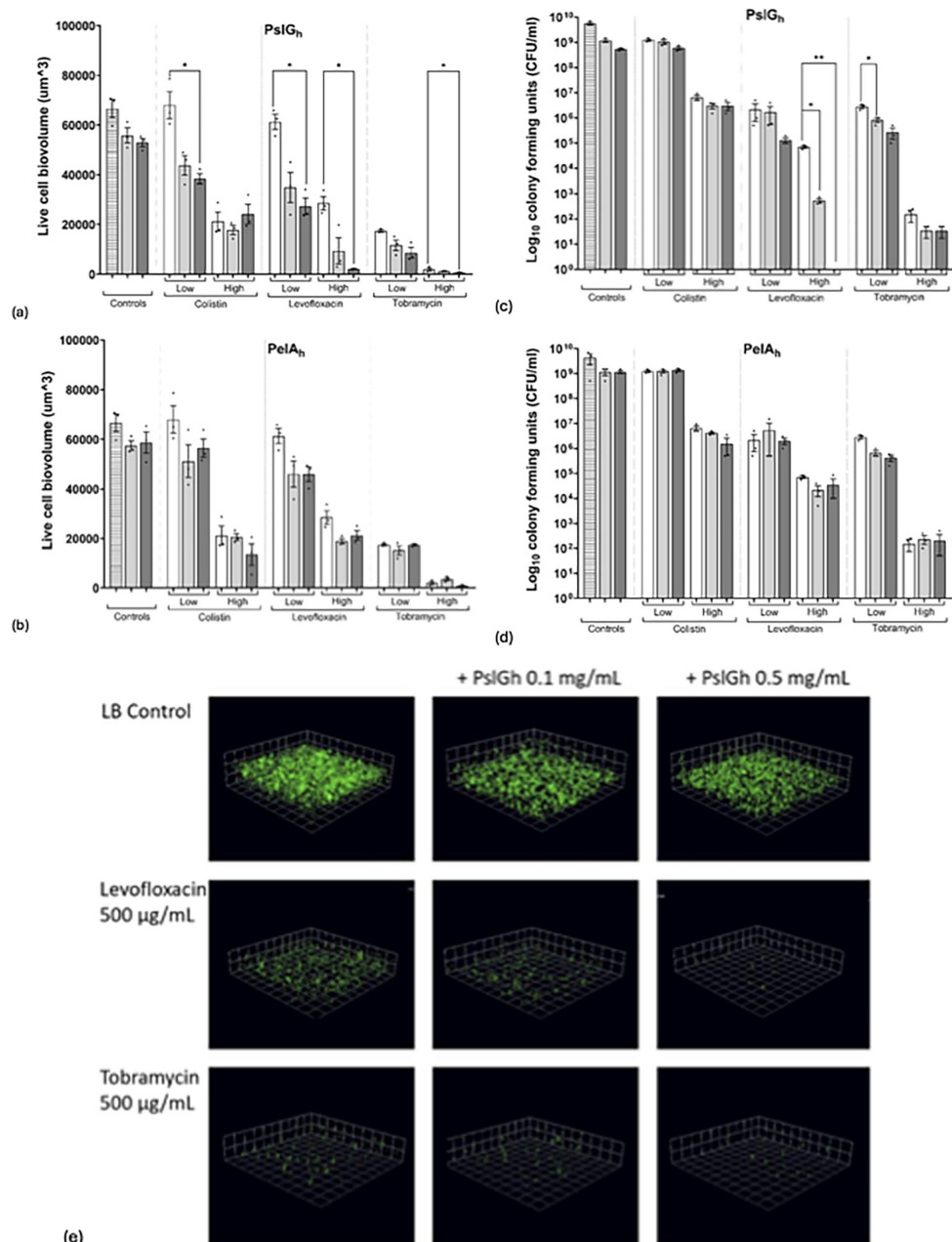

**FIG 3** Confocal microscopy of PA342 biofilms treated with antibiotics and GHs. (a and b) Live-cell biovolume following treatment with antibiotics plus PslG$_h$ (a) or PelA$_h$ (b). (c and d) CFU/mL for PslG$_h$ (c) or PelA$_h$ (d). (e) Representative confocal images showing biovolume reduction with levofloxacin or tobramycin and PslG$_h$. Bars represent means ± SEM of three biological replicates. Four technical replicates were used for biovolume; three for CFU counts. Striped bars indicate LB alone, white = antibiotic alone, light gray = low (0.1 mg/mL), and dark gray = high (0.5 mg/mL) GH concentration. Antibiotic concentrations: colistin (8, 40 µg/mL), levofloxacin (100, 500 µg/mL), and tobramycin (100, 500 µg/mL). Statistical comparisons were performed by one-way ANOVA with post hoc testing versus antibiotic alone. *$P < 0.05$; **$P < 0.01$.

## PelA$_h$-disrupted strain PA380 responds to PelA$_h$, but not PslG$_h$

PA380 was tested in parallel using the same antibiotic and GH combinations (Fig. 4).

PelA$_h$, but not PslG$_h$, was associated with statistically significant reductions in biofilm biovolume and/or viable counts in combination with levofloxacin, tobramycin, and colistin under specific concentration conditions. For levofloxacin, reductions in biovolume were seen at both antibiotic doses and both enzyme doses (Fig. 4b), and

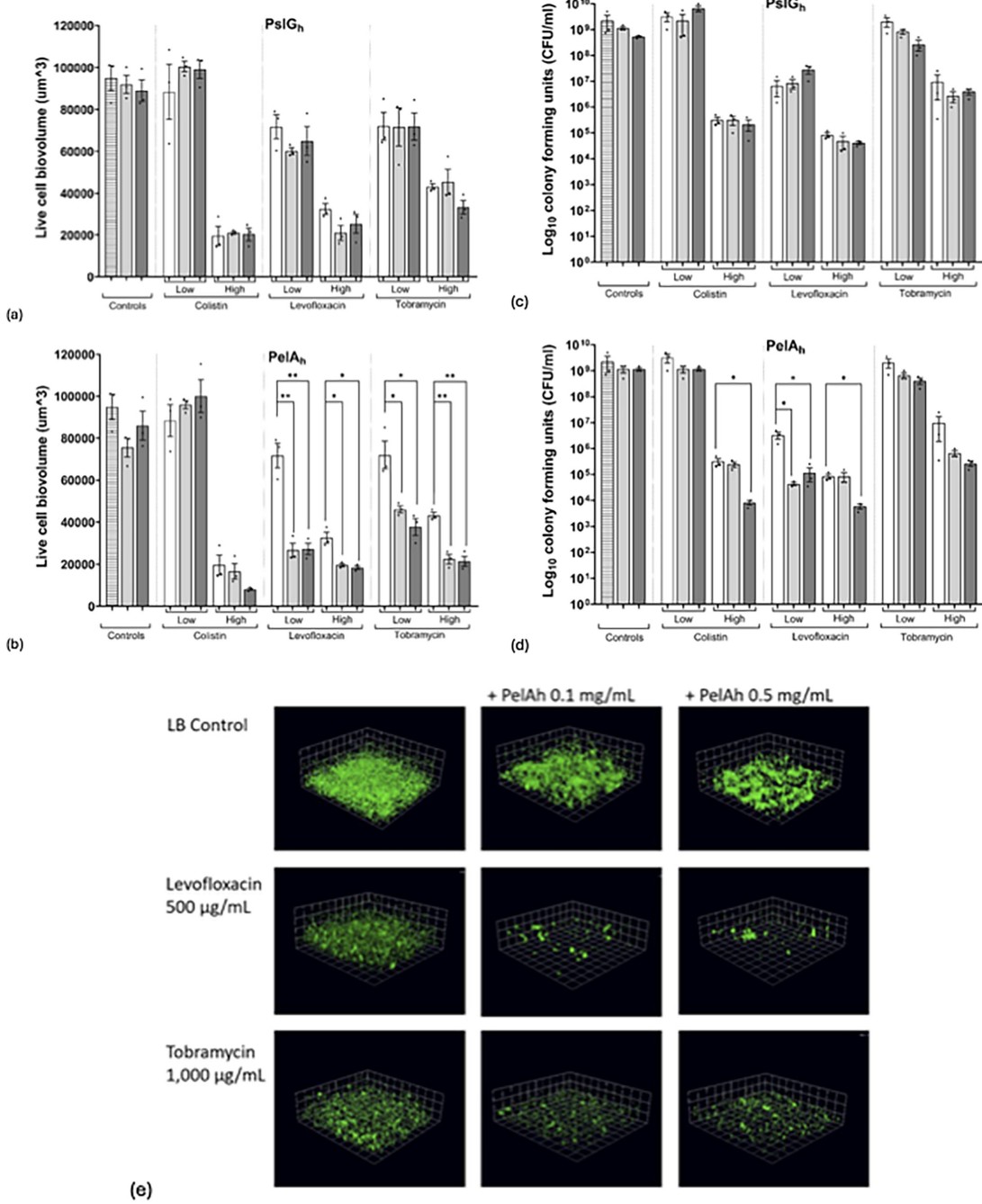

**FIG 4** Confocal microscopy of PA380 biofilms treated with antibiotics and GHs. (a and b) Live-cell biovolume with antibiotic plus PslG$_h$(a) or PelA$_h$(b). (c and d) CFU/mL for PslG$_h$(c) or PelA$_h$ (d). (e) Representative images showing biovolume reduction with levofloxacin or tobramycin and PelA$_h$. Bars represent means ± SEM of three biological replicates. Four technical replicates were used for biovolume; three for CFU counts. Striped bars indicate LB alone, white = antibiotic alone, light gray = low (0.1 mg/mL), and dark gray = high (0.5 mg/mL) GH concentration. Antibiotic concentrations: colistin (8, 40 µg/mL), levofloxacin (100, 500 µg/mL), and tobramycin (200, 1000 µg/mL). Statistical comparisons were performed by one-way ANOVA with post hoc testing versus antibiotic alone. *$P < 0.05$; **$P < 0.01$.

this was seen in the CFU quantification only with the high dose of PelA$_h$ (Fig. 4d). For tobramycin, significant biovolume reductions were observed with both PelA$_h$ doses (Fig. 4b), although CFU reductions did not reach statistical significance, despite downward trends (Fig. 4d). Biomass quantification via live-cell staining did not reveal a dose response, as both low and high concentrations of PelA$_h$ appeared similarly effective.

However, CFU quantification showed additive effects primarily at the higher $PelA_h$ concentration (three instances of significant reduction at the higher GH concentration versus only one at the lower GH concentration), raising the possibility of a concentration-dependent trend that was not observed with biovolume quantification.

## Differential susceptibility of $PslG_h$ and $PelA_h$ to degradation by CF sputum supernatant

Although both $PslG_h$ and $PelA_h$ rapidly degrade their respective matrix exopolysaccharides (5–15 min) (25–27), antibiotic action occurs over a longer timeframe, particularly in biofilms (hours to days) (39). In clinical settings, inhaled antibiotics are typically administered two to three times daily, raising the question of whether co-administered enzymatic adjuncts retain detectable activity within the protease-rich CF airway environment.

To evaluate susceptibility to degradation under these conditions, $PslG_h$ and $PelA_h$ were incubated at 37 °C in LB with or without sputum supernatant (10% and 20%) for 0, 4, and 20 h, and protein integrity was assessed by immunoblotting (Fig. 5). $PslG_h$ and $PelA_h$ have predicted molecular weights of 45–50 kDa and 20–25 kDa, respectively (40).

$PslG_h$ remained detectable at all time points in LB alone, with almost full degradation observed at 20 h (faint band in Fig. 5a) when incubated with CF sputum supernatant. In contrast, while $PelA_h$ was stable in LB, with some degradation seen at the 4- and 20-hour time points, no detectable $PelA_h$ bands remained at 4 or 20 h in the presence of either 10% or 20% sputum supernatant (Fig. 5b). These results indicate that while both enzymes are susceptible to degradation in CF airway conditions, $PslG_h$ demonstrates greater stability than $PelA_h$, supporting its potential utility as a more robust therapeutic adjunct.

To determine whether residual enzymatic activity could still be detected following exposure to CF sputum supernatant under the same conditions used for immunoblotting, we performed a functional biofilm disruption assay using laboratory strains with well-defined matrix architectures (PAO1 for $PslG_h$ and PA14 for $PelA_h$). Following pre-incubation, $PslG_h$-treated biofilms exhibited measurable disruption at early time points (0 and 4 h) in both buffer and sputum-containing conditions, with reduced effects observed only after prolonged incubation (20 h), including in the absence of sputum supernatant. In contrast, $PelA_h$-mediated effects were diminished in sputum-containing conditions and were not consistently detectable at later time points. Results are presented descriptively in Table S3 at https://zenodo.org/records/18701433.

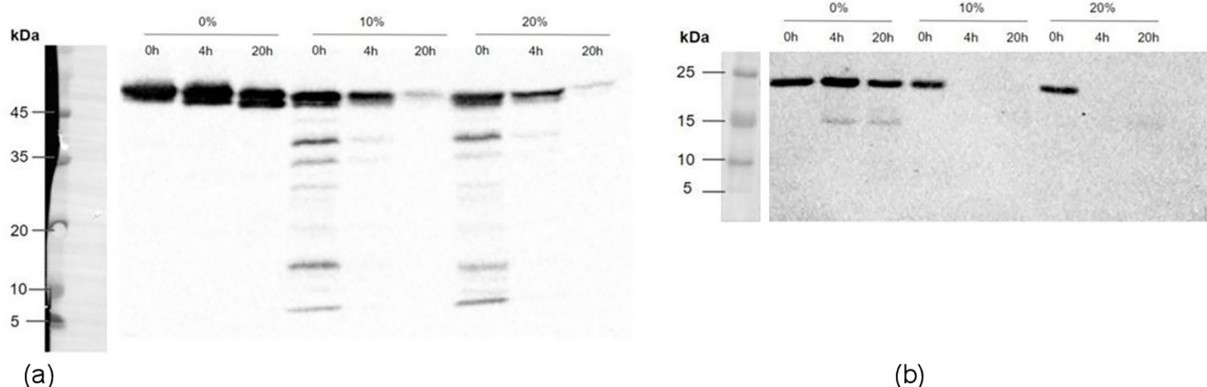

**FIG 5** Western blots for $PslG_h$ and $PelA_h$ using protein-specific antibodies, incubated at time points with and without sputum supernatant. (a) Western blot for $PslG_h$ (45–50 kDa) at 0-, 4-, and 20-hour time points individually and with 10% and 20% sputum supernatant. (b) Western blot for $PelA_h$ (20–25 kDa) at 0-, 4-, and 20-hour time points individually and with 10% and 20% sputum supernatant.

## DISCUSSION

*P. aeruginosa* lung infection in CF is associated with worse clinical outcomes despite current antimicrobial therapies, highlighting the need for novel adjunctive treatments. In this study, we sought to first quantify the disruption of *P. aeruginosa* biofilms with application of GH targeting Psl and Pel across a genetically distinct set of early CF clinical isolates, before evaluating their potential as antibiotic adjuvants. Our dosing strategy (0.05–0.1 mg/mL) mirrors concentrations shown to be effective *in vitro* (25) and in murine infection models (26, 27), supporting the translational relevance of the GH activities observed in our experiments.

In our functional assay, we demonstrate that early *P. aeruginosa* CF isolates exhibit functional dependence on both Psl and Pel, with disruption by $PslG_h$ predominating. Among 14 early-infection isolates, a larger proportion were disrupted by $PslG_h$ than by $PelA_h$. While fewer strains were $PelA_h$-disrupted, this GH still showed significant adjuvant activity—particularly in strains where $PslG_h$ + antibiotic combinations were ineffective—highlighting its potential value in select cases. Considering the limited accuracy of the biofilm quantification approaches employed, some of the significant changes observed may fall within the range of experimental errors. Also, given the inherent variability of biofilm quantification assays, effect sizes were interpreted in conjunction with statistical significance, antibiotic killing assays, and strain-specific structural biofilm disruption observed by confocal microscopy.

In high-throughput assays, both GHs enhanced antibiotic activity when combined with antibiotics from different mechanistic classes. We considered whether physicochemical interactions—particularly charge, polarity, and matrix composition—might affect GH-antibiotic additivity. Tobramycin and $PelA_h$ are both positively charged, while colistin (as a sulfate salt) carries a net negative charge upon dissociation. Alginate, a third major *P. aeruginosa* exopolysaccharide that predominates in mucoid phenotypes, is also highly anionic due to its uronic acid backbone (41). These charge differences may influence antibiotic diffusion, matrix binding, or intracellular uptake. Notably, GH-colistin combinations exhibited lower adjunctive effects in our experiments. This may reflect reduced colistin concentrations in airway secretions due to its large molecular size and poor sputum bioavailability compared to levofloxacin or tobramycin (36). Alternatively, electrostatic repulsion between positively charged compounds and cationic matrix components, such as Pel and DNA, may impede effective binding or diffusion.

Beyond charge effects, enzymatic degradation of the biofilm matrix may also alter biophysical properties that govern antibiotic penetration, including matrix porosity, density, and hydration. Prior work has shown that enzymatic disruption of biofilms can increase diffusivity by reducing matrix crosslinking and compactness (42–44). By cleaving specific polysaccharides (Psl or Pel), GHs may loosen the biofilm network, reduce steric hindrance, or modulate the binding affinity of matrix components for antibiotics. These physicochemical changes may partially account for the enhanced antibiotic susceptibility observed with GH co-treatment. Although we did not measure matrix structure or permeability directly, our confocal imaging provides qualitative support for altered biofilm integrity following GH exposure. Future studies using high-resolution imaging, microviscosity measurements, or charge-mapping techniques could more precisely define these biophysical effects and help optimize GH-antibiotic pairings.

We acknowledge that the static 96-well plate biofilm model and crystal violet staining primarily assess surface-attached biomass and do not fully replicate the complex, dynamic conditions of chronic biofilm infections in the CF lung. However, this model allows for high-throughput screening across multiple clinical isolates. As a complementary approach, we employed live-cell staining with confocal microscopy in select strains to provide additional insight into biofilm architecture and cell viability. This additional method provided further validation, showing significant enhancement of antibiotic efficacy at subinhibitory concentrations in strains disrupted by either $PslG_h$ or $PelA_h$ when the corresponding GH was used. No additive effect was observed when the GH targeted an EPS not associated with biofilm susceptibility in the screening

assay, even at higher concentrations, suggesting selective matrix disruption. A clear dose-response relationship was not consistently observed, raising questions about biofilm accessibility or whether target saturation occurs at lower GH concentrations. This finding supports the concept that targeting the exopolysaccharide most responsive to enzymatic disruption may be critical, reinforcing the rationale for phenotypically guided, strain-specific adjuvant therapy. Rapid diagnostics such as the high-throughput assay employed in this study could enable personalized adjunctive therapy with the appropriate GH, potentially improving treatment outcomes in CF.

This study focused on first *P. aeruginosa* isolates from pediatric patients—seven that were successfully eradicated and seven that persisted despite antibiotic therapy, causing chronic infection—and the findings cannot be directly extrapolated to chronic or MDR isolates, which may exhibit different biofilm matrix profiles. Future work should assess exopolysaccharide expression patterns in a broader spectrum of isolates including those with known virulence factors or antimicrobial resistance, ideally using transcriptomic or proteomic approaches, to determine whether targeted GH therapies remain effective over time or require adaptation.

These results have implications for potential clinical use: either administering both GHs simultaneously or selectively deploying the relevant GH following diagnostic screening for Psl and Pel expression. However, while dual GH treatment was tested in two representative isolates, we chose not to generalize these findings across all strains, as the selected isolates were intentionally chosen for their distinct responses to individual GHs. It remains possible that additive or synergistic effects could emerge in strains with mixed or ambiguous matrix phenotypes, and broader testing would be needed to assess this. Indeed, previous work has shown additive effects when both are applied concurrently in a limited number of clinical strains (27). Future studies should test dual GH therapy more broadly and assess whether it can overcome limitations observed with monotherapy. Nebulized formulations incorporating dual GHs could provide broad-spectrum activity against matrix-dense biofilms, even when the biofilm exopolysaccharide composition is unknown or heterogeneous, offering a practical approach for empiric use.

Protein stability is a critical consideration in clinical development. Both enzymes were stable at body temperature in buffer, as demonstrated by Western blot. However, PelA$_h$ underwent significant degradation, especially in the presence of CF sputum supernatant, with complete loss of detectable protein at 4 and 20 h. This is certainly a factor that might be a key translational challenge of this GH to clinical use. The rapid degradation seen in our experiments suggests the presence of sputum proteases or other degradative factors that may limit PelA$_h$'s *in vivo* utility. While the specific enzymes responsible for PelA$_h$ degradation remain to be identified, candidates include neutrophil elastase, cathepsins, and other serine or metalloproteases enriched in CF airways. Strategies to improve enzyme half-life may include encapsulation in protective nanoparticles or exploring the use of more protease-resistant orthologs from other species. These approaches could enhance therapeutic potential and reduce dosing frequency in clinical settings.

## Conclusion

PslG$_h$ and PelA$_h$ are promising adjuvant therapies for enhancing antibiotic activity against *P. aeruginosa* biofilms in CF. Our findings demonstrate variable reliance on Psl and Pel among early clinical isolates, supporting the exploration of tailored therapies. The ability of these enzymes to potentiate antibiotic activity—especially at clinically relevant concentrations—reinforces their translational potential. However, further development will require improved understanding of their performance in MDR infections, optimized delivery strategies, and enhanced stability in sputum-rich environments.

## ACKNOWLEDGMENTS

We would like to acknowledge the Microbiology Department at The Hospital for Sick Children for helping us biobank sputum specimens from patients with CF. We would also like to acknowledge the families who consented for their children being enrolled in the sputum biobank at our institution.

## AUTHOR AFFILIATIONS

[1]Division of Paediatric Respiratory Medicine, Department of Paediatrics, The Hospital for Sick Children, Toronto, Ontario, Canada

[2]Program in Translational Medicine, Peter Gilgan Center for Research and Learning, The Hospital for Sick Children, Toronto, Ontario, Canada

[3]Temerty Faculty of Medicine, University of Toronto, Toronto, Ontario, Canada

[4]Program in Molecular Medicine, Peter Gilgan Center for Research and Learning, The Hospital for Sick Children, Toronto, Ontario, Canada

[5]Department of Cellular and Molecular Biology, University of Texas Health Science Center at Tyler, Tyler, Texas, USA

[6]Department of Biochemistry, University of Toronto, Toronto, Ontario, Canada

[7]Division of Infectious Diseases, Department of Paediatrics, The Hospital for Sick Children, Toronto, Ontario, Canada

## AUTHOR ORCIDs

Isaac Martin  http://orcid.org/0000-0003-4886-189X
Amanda J. Morris  http://orcid.org/0000-0003-3490-0691
Shafinaz Eisha  http://orcid.org/0000-0001-5537-6249
P. Lynne Howell  http://orcid.org/0000-0002-2776-062X
Valerie Waters  http://orcid.org/0000-0003-1566-6510

## FUNDING

| Funder | Grant(s) | Author(s) |
|---|---|---|
| Canadian Institutes of Health Research | FDN154327 | P. Lynne Howell |
| GlycoNET | ID-03 | P. Lynne Howell |
| | | Valerie Waters |
| GlycoNET | ID-11 | Valerie Waters |

## AUTHOR CONTRIBUTIONS

Isaac Martin, Conceptualization, Data curation, Formal analysis, Investigation, Methodology, Validation, Visualization, Writing – original draft, Writing – review and editing | Jonathan Chung, Investigation, Methodology, Project administration | Deepa Raju, Formal analysis | Amanda J. Morris, Methodology | Shafinaz Eisha, Methodology | Valerie Waters, Conceptualization, Formal analysis, Project administration, Supervision.

## DATA AVAILABILITY

All whole -genome sequenced data from these isolates are available through Bioproject PRJNA556419 with the following GenbankGenBank accessions: JAGHM W000000000 (Pa50), JAGHMV000000000 (Pa263), JAGHMU000000000 (Pa288), JAGHMT000000000 (Pa325), JAGHMS000000000 (Pa342), JAGHM R000000000 (Pa375), JAGHMQ000000000 (Pa380), JAGHMP000000000 (Pa404), JAGHMO000000000 (Pa505), JAGHMN000000000 (Pa551), JAGHM M000000000 (Pa558), JAGHML000000000 (Pa565), JAGHMK000000000 (Pa580), and JAGHMI000000000 (Pa549).

## ADDITIONAL FILES

The following material is available online.

## Open Peer Review

**PEER REVIEW HISTORY (review-history.pdf).** An accounting of the reviewer comments and feedback.

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
