## [Reviewer comments · Microbiology Spectrum]

Microbiology Spectrum

Glycoside Hydrolases Enhance Antibiotic Activity Against *Pseudomonas aeruginosa* Biofilms from Cystic Fibrosis Airways

Isaac Martin, Jonathan Chung, Deepa Raju, Yvonne Yau, Amanda Morris, Shafinaz Eisha, P. Howell, and Valerie Waters

Corresponding Author(s): Isaac Martin, SickKids Research Institute

Review Timeline:

Submission Date:	April 29, 2025
Editorial Decision:	June 11, 2025
Revision Received:	August 9, 2025
Editorial Decision:	October 21, 2025
Revision Received:	December 3, 2025
Editorial Decision:	December 17, 2025
Revision Received:	December 23, 2025
Accepted:	January 10, 2026

Editor: Marianna Patrauchan

Reviewer(s): Disclosure of reviewer identity is with reference to reviewer comments included in decision letter(s). The following individuals involved in review of your submission have agreed to reveal their identity: Michael D. Parkins (Reviewer #1)

Transaction Report:

DOI: <https://doi.org/10.1128/spectrum.01354-25>

Re: Spectrum01354-25 (Breaking Biofilms: Enhancing Antibiotic Efficacy with Glycoside Hydrolases in *Pseudomonas aeruginosa* from Cystic Fibrosis Airways)

Dear Dr. Isaac Martin:

Thank you for the privilege of reviewing your work. Below you will find my comments, instructions from the Spectrum editorial office, and the reviewer comments.

Please address all reviewers' comments thoroughly, with particular attention to the limited number of strains, concentrations of antibiotics, and data presentation. The data presented are insufficient to define the strains as Pel- or Psl-dominant. All suggested revisions, including additional experiments, are needed for the manuscript to be considered for publication in Microbiology spectrum.

Revision Guidelines

Sincerely,
Marianna Patrauchan
Editor
Microbiology Spectrum

Reviewer #1 (Comments for the Author):

Martin et al. present a manuscript which explores glycoside hydrolases as a means of improving biofilm treatment/dispersion of

P. aeruginosa infections - as may be relevant to those with cystic fibrosis.

The manuscript is generally well written - and simple, easy to follow.

There are a few modest suggestions that I can make - but there is nothing obvious that would suggest that this work should not be considered for publication in *Spectrum* - it is quite good.

More significant suggestions

1. Isolates chosen

- a. Include as a limitation the choice of early PA isolates from individuals with CF - which may or may not have the attributes of those causing chronic infection.
- b. Information on if these are related strains, come from different persons should be included
- c. Information on MIC of each isolate to the three target Abx is relevant given the use of standard Abx concentrations which may not have the same effect on different strains (ie for one strain the Abx concentration used might be 10X that of the isolates MIC - and another it might 1/10th - and different outcomes would reasonably be expected). As early isolates one might presume these to be fairly susceptible - but this should be established.

2. Methods. Please clarify the subinhibitory concentrations of antibiotics - as this would be strain dependent. Terminology for low and high concentrations explains the rationale (great!) - but would be nice to have exact values as it was hard to follow (and in the figure low/high - at least the legend)

3. Results - Presentation of results in this section as proportion of reduction could be a bit misleading - as it does not include information on how much mass was present to begin with, and whether some isolates are better "biofilm" formers than others. This would be good to include (ie absolute vs relative reduction)

4. Discussion - Limitations - the biofilm model is merely assessing static adhesion using 96 well plates with biomass being assessed by the Crystal violet assay which may be less relevant to chronic infections in the CF lung - and this should be commented on. A strength is the use of the confocal microscopy for a couple of the isolates which distinguished live cells.

5. asdf

Minor suggestions

1. Abstract

- a. Peak sputum concentrations "following nebulization" - or something else to imply airways delivery
- b. Conclusions - support "exploration of" tailored strategies

2. Introduction

a. Last paragraph summarizes the findings - and this could be more abstractable - and here goals of the project could be presented. (this is a very minor point - please disregard as it is your study...)

3. Methods

a. Sub inhibitory concentrations of antibiotics - dose is not clear (but is stated in results), nor is the meaning given everything above the planktonic MIC would not qualify with this term.

4. Results

a. Figure 3 conditions are listed as low and high (subjective terms)... But the text and figure have different values in them - and would be more accurate to list actual values for each (especially as we have very low sub-inhibitory values earlier).

5. Discussion

a. Inhaled antibiotics would not be considered bioavailable as topically delivered

Reviewer #2 (Comments for the Author):

The manuscript "Breaking Biofilms: Enhancing Antibiotic Efficacy with Glycoside Hydrolases in *Pseudomonas aeruginosa* from Cystic Fibrosis Airways" by Martin and coworkers describes the effects of glycosyl hydrolases on biofilms of cystic fibrosis clinical isolates. The authors also attempt to determine if the GHs potentiate antibiotic penetration into *P. aeruginosa* biofilms. However, the concentrations of antibiotics used in this study were not valid to give relevant information. The title overstates the effects of the GHs on disruption of the biofilms in the microtiter plate model. The title also gives the indication that this is a review article rather than a primary research article.

There are many major flaws with this manuscript, most having to do with using the wrong antibiotic concentrations, not saying what concentrations were used in some experiments. There is a lack of direct chemical assays showing the types of polysaccharides produced by these isolates. Some figures lack of figure legends, which makes it even more difficult to figure out what was done in the experiments.

Other minor flaws in the written text are noted below.

Abstract:

The abstract is not formatted as should be for Microbiology Spectrum articles.

Introduction:

Pg 2.

Ln 38 - delete "significant"

Ln 14 - I don't think that's the proper use of the word "adjuvant". Also need a citation for this sentence.

Ln 15 - write out what GH is when it is used first.

Ln 16 - I assume alginate producing strains are also found in CF lungs. Those aren't mentioned in the introduction.

Pg 3.

Ln 1. Define what enzymes PslGh and PelAh are when used for the first time.

Ln 2. Ref 23, 24 - What in vivo model was used to test activity?

Ln 3 - reference?

Ln 4 - what wound infection model?

Ln 12 - what do you mean by "more than". Vague.

Ln 15 - Confocal Microscopy doesn't confirm anything, the experiment does. What was the experiment?

Ln 16 - also a vague statement, what do you mean by "stable".

The Introduction has many very vague statements. I think that it should be re-written to clarify the statements and make a stronger introduction.

Methods:

Pg 3

Ln 20 - What is an "early" isolate?

Pg 5

Ln 8 - "Two subinhibitory antibiotic concentrations". What were the concentrations?

Ln 13 - only using two concentrations doesn't assess dose response. You would need a standard curve for that.

Results

Figure 1 - You need some sort of assay (chemical or spectroscopic - something other than GH sensitivity) to define what polysaccharides are produced by these strains. GHs may have off target effects. In order to have a direct cause and effect result, you need to know what polysaccharide(s) each of those strains are producing.

Figure 1 B - since the different strains are likely producing different polysaccharides, this figure is (based on percentage) not very meaningful.

Figure 2 - The concentrations of antibiotics used here are too high to obtain valid results in vitro, and far exceed what might be used in vivo. In fact, I was curious if the antibiotics were even soluble at those concentrations. The solubility of levofloxacin is 54 ug/ml.

<https://pubchem.ncbi.nlm.nih.gov/compound/Levofloxacin#section=Melting-Point>

So, there is no way that levofloxacin (4,500 µg/mL) was used. Either the math was done incorrectly, or the amount of antibiotic used was unreasonable.

In either case, this manuscript should not have been submitted for publication as is.

The figures that don't have figure legends, refer to "low" and "high" concentrations of the antibiotics, but there is no figure legend that says what those concentrations are. I assume this goes back to the vague description in the methods section (which also doesn't state the concentrations).

Figure 4 - Similarly, the concentrations of antibiotics used in these experiments is not reasonable. The y-axis says "500 mg/ml". I assume the authors meant to say "500 ug/ml", but even if that's the case, that is above the solubility for levofloxacin, making this figure irrelevant to the effects of antibiotics.

Reviewer #3 (Comments for the Author):

The manuscript by Martin et al. reports a study on the glycoside hydrolases (GHs) PslGh and PelAh and their ability to enhance antibiotic efficacy against *Pseudomonas aeruginosa* biofilms. Fourteen clinical isolates from pediatric cystic fibrosis (CF) patients were tested, using both crystal violet staining and confocal microscopy to evaluate treated biofilms. The study is relevant to the field and is methodologically sound overall. However, several weaknesses should be addressed.

The use of only 14 isolates is a limitation, particularly given the authors' aim to compare GH activities "across diverse clinical isolates." A larger number of strains would strengthen the conclusions about differential GH activity against Psl and Pel, especially as some strains may produce minimal Pel EPS. It would also be valuable to quantify the EPS content in the tested biofilms.

The authors reference prior work on dual GH treatments; it would strengthen the paper to include at least a subset experiment testing combined PslGh + PelAh on a representative strain. Additionally, alginate - another major EPS component in *Pseudomonas* biofilms - should be at least acknowledged and discussed in the manuscript.

The mechanisms underlying the observed effects should be explored in greater depth. The interactions between antibiotics (e.g., colistin) and specific matrix components could be elaborated, as could the discussion of physicochemical alterations to the biofilm post-GH treatment (e.g., changes in porosity or charge repulsion).

Minor issues:

In Figure 2, consider including raw biomass values alongside percentage reductions for clearer context.

Some text could be refined, such as the Abstract, line 26: "yielding up to 18.1% further biofilm reduction" - consider rephrasing for clarity.

On page 2, line 15, when GH is first mentioned, the acronym should be defined.

Please clarify how the working concentrations of 0.1 mg/mL for crystal violet assays and 0.1 or 0.5 mg/mL for confocal experiments were selected.

The titles in line 14 on page 11 and line 19 on page 14 appear to be subtitles under the main title on line 3 of page 11 and should be formatted accordingly.

The authors may also wish to reference Secor PR et al., 2018 (Matrix penetration by antibiotics in sputum biofilms).

**Manuscript
Number**

Manuscript Title

Spectrum01354-25

Breaking Biofilms: Enhancing Antibiotic Efficacy with Glycoside Hydrolases in *Pseudomonas aeruginosa* from Cystic Fibrosis Airways

Please note that all changes reference the Marked Up Manuscript – No Figures document

EDITOR/REVIEWER COMMENTS The specific editor and reviewer queries.	AUTHOR'S RESPONSE Response to the editor and reviewer queries here.	REFERENCE PAGE AND LINE Where the change appears in the revised manuscript.
Reviewer 1: Include as a limitation the choice of early PA isolates from individuals with CF - which may or may not have the attributes of those causing chronic infection.	We thank the reviewer for this comment. We have added to a small paragraph in the discussion section to better highlight this point. "This study focused on first P. aeruginosa isolates from pediatric patients - 7 that went on to be eradicated and 7 that persisted despite antibiotic therapy, causing chronic infection - and the findings cannot be directly extrapolated to chronic or MDR isolates, which may exhibit different biofilm matrix profiles. Future work should assess exopolysaccharide expression patterns in a broader spectrum of isolates including those with known virulence factors or antimicrobial resistance, ideally using transcriptomic or proteomic approaches, to determine whether targeted GH therapies remain effective over time or require adaptation."	Page 14. Lines 11-17.
Reviewer 1: Information on if these are related strains, come from different persons should be included.	This is a valid point and we have now included a supplementary Table 1, which has been uploaded in the Supplementary Materials document. We have added the following sentence within the manuscript to point the reader in the direction of our Supplementary Materials document: "The 14 clinical P. aeruginosa isolates used in this study were from different paediatric patients at our centre and the clinical and demographic characteristics of these patients are detailed in Supplemental Table 1." Additionally,	Page 7. Lines 12-14. Supplementary Materials Table 1.
Reviewer 1: Information on MIC of each isolate to the three target Abx is relevant given the use of standard Abx concentrations which may not have the same effect on different strains (ie for one strain the Abx concentration used might be 10X that of the isolates MIC - and another it might 1/10th - and different outcomes would reasonably	We thank the reviewer for this comment. We have now included a supplementary Table 2, which has been uploaded in the Supplementary Materials document. We have added the following sentence within the manuscript to point the reader in the direction of our Supplementary Materials document: "Minimum inhibitory concentration (MIC) values for each isolate to the three antibiotics tested are listed in Supplementary Table 2."	Page 7. Lines 14-15.

be expected).		
Reviewer 1: Methods. Please clarify the subinhibitory concentrations of antibiotics - as this would be strain dependent. Terminology for low and high concentrations explains the rationale (great!) - but would be nice to have exact values as it was hard to follow (and in the figure low/high - at least the legend).	We thank you for bringing our attention to this potential ambiguity. We have now given these exact values in both the figure legends and in the methods section. The expanded methods section now reads: “Two subinhibitory antibiotic concentrations were selected for each agent. To assess the effects of glycoside hydrolase–antibiotic combinations on biofilm architecture and cell viability, we selected strain-specific “high” and “low” antibiotic concentrations for confocal microscopy experiments. For each isolate, we began by testing antibiotics at the peak sputum concentrations used in the previous experiments. If no viable biofilm structure or live cells were observed by confocal microscopy with live/dead staining at this concentration, we performed serial twofold dilutions until a viable biofilm was detectable. The highest concentration at which a viable biofilm could still be observed was designated the “high” concentration. The “low” concentration was defined as 20% of this high value. For isolate PA342, the low and high antibiotic concentrations (µg/mL) were: colistin (8, 40), levofloxacin (100, 500), and tobramycin (100, 500). For isolate PA380: colistin (8, 40), levofloxacin (100, 500), and tobramycin (200, 1000). In cases where biofilm viability was still present at the initial peak sputum concentration, this value was taken as the “high” concentration, and the “low” was set at 20% of that.”	Page 5. Lines 15-24.
Reviewer 1: Results - Presentation of results in this section as proportion of reduction could be a bit misleading - as it does not include information on how much mass was present to begin with, and whether some isolates are better "biofilm" formers than others. This would be good to include (ie absolute vs relative reduction).	We thank the reviewer for this helpful suggestion. We agree that reporting only the relative percent reduction in biofilm biomass may obscure the variation in baseline biofilm-forming capacity between isolates. Our primary motivation for using relative reduction was to allow cross-strain comparisons in a standardized format, particularly given the large variability in absolute biofilm biomass and the challenge of graphically presenting all 14 isolates with different baseline OD ranges. To address the reviewer’s concern, we have now included Supplementary Figure 1, which graphs the average OD₆₀₀ values of untreated negative control wells for each of the 14 isolates alongside those for the GHs. This provides transparency regarding baseline biofilm biomass and allows readers to contextualize the magnitude of biofilm reduction observed. We have also added a corresponding sentence in the Results section and updated the figure legends to point readers to Supplementary Figure 1: “To facilitate comparison across strains, biofilm biomass reduction is presented as a percentage relative to the untreated control for each isolate. However, as baseline biofilm formation varied between isolates, the average OD₆₀₀ values of the negative control wells for all 14 strains are provided in Supplementary Figure 1.”	Page 8. Lines 1-3. Supplementary Materials: Supplementary Figure 1.
Reviewer 1: Discussion - Limitations - the biofilm model is merely assessing	We thank the reviewer for this comment. We have added the following in our Discussion section: “We acknowledge that the static 96-well plate biofilm model and	Page 13. Lines 20-24.

static adhesion using 96 well plates with biomass being assessed by the Crystal violet assay which may be less relevant to chronic infections in the CF lung - and this should be commented on. A strength is the use of the confocal microscopy for a couple of the isolates which distinguished lives cells.	crystal violet staining primarily assess surface-attached biomass and do not fully replicate the complex, dynamic conditions of chronic biofilm infections in the CF lung. However, this model allows for high-throughput screening across multiple clinical isolates. As a complementary approach, we employed live cell staining with confocal microscopy in select strains to provide additional insight into biofilm architecture and cell viability.”	
Reviewer 2: The concentrations of antibiotics used in this study were not valid to give relevant information.	We thank the reviewer for this comment. We have chosen 3 mechanistically different antibiotics, all of which are used in clinical practice via nebulization. The antibiotic concentrations were carefully selected based on mean peak sputum concentrations that are achievable through nebulization. We recognize that these antibiotic concentrations are often many orders of magnitude higher than those that be achieved via other routes of administration (oral or intravenous). There is a good reference that we have used (Chmiel et al., Annals ATS, 2014) that summarizes these values and references specific studies where these concentrations have been calculated. We have expanded the sentence in the methods section that summarizes this approach: “Final working concentrations (based on reported peak sputum levels via nebulization in CF patients) were: colistin 40 µg/mL, levofloxacin 4,500 µg/mL, and tobramycin 1,000 µg/mL (31-33).”	Page 4. Lines 19-20.
Reviewer 2: The title overstates the effects of the GHs on disruption of the biofilms in the microtiter plate model. The title also gives the indication that this is a review article rather than a primary research article.	We thank the reviewer for this comment. The intention of the title was to reflect the observed enhancement of antibiotic efficacy when combined with glycoside hydrolases, rather than to imply complete biofilm eradication. We acknowledge that “Breaking Biofilms” may be interpreted as overstating the extent of disruption in a microtiter plate model. To avoid any potential overstatement, we have now changed the title to “Glycoside Hydrolases Enhance Antibiotic Activity Against Pseudomonas aeruginosa Biofilms from Cystic Fibrosis Airways”.	Page 1. Lines 1-2.
Reviewer 2: There is a lack of direct chemical assays showing the types of polysaccharides produced by these isolates.	We thank the reviewer for this insightful comment. We agree that direct chemical assays of exopolysaccharides would provide valuable mechanistic information about the biofilm matrix composition in these isolates. However, the objective of our study was to develop a functional screening approach that could be applied to a large number of clinical isolates, similar to what might be implemented in a diagnostic or therapeutic stratification context. Given the diversity of biofilm matrix components across P. aeruginosa isolates and the resource-intensive nature of chemical characterization, we instead used glycoside hydrolase susceptibility as a phenotypic proxy to infer dominant exopolysaccharide usage. We have clarified this rationale in the revised manuscript in our methods section to avoid confusion about the scope of the study. “This approach was chosen	Page 5. Lines 5-8.

	to allow high-throughput screening across multiple clinical isolates. Direct chemical analysis of matrix polysaccharides was not performed due to its complexity and limited feasibility in translational or clinical applications.”	
Reviewer 2: Some figures lack of figure legends, which makes it even more difficult to figure out what was done in the experiments.	All of our figures have figure legends. We apologize if any of the Figure legends were not visible in the previous manuscript. Please let us know if this is not the case, and we can try to reformat the Figures and Legends.	
Reviewer 2: The concentrations of antibiotics used here are too high to obtain valid results in vitro, and far exceed what might be used in vivo. In fact, I was curious if the antibiotics were even soluble at those concentrations. The solubility of levofloxacin is 54 ug/ml.	We appreciate the reviewer’s concern regarding the high concentrations of antibiotics used in our experiments. However, these concentrations were intentionally selected based on reported peak sputum levels achieved in individuals with cystic fibrosis receiving standard dosing regimens. In the case of levofloxacin, for example, mean sputum concentrations exceeding 4,000 µg/mL have been reported following inhaled administration. Our goal was to simulate relevant local antibiotic exposure within the biofilm-infected airways, rather than systemic levels achieved by oral or intravenous administration. Regarding solubility, we agree that aqueous solubility of levofloxacin at neutral pH is limited (~54 µg/mL in water); however, we prepared concentrated stock solutions by adjusting pH, as per the manufacturer protocol, with a final concentration of 100 mg/mL. Levofloxacin solubility can also be further increased with alternative solvents such as DMSO. All antibiotics were fully dissolved before use. We have expanded the sentence in the methods section that summarizes this approach: “Final working concentrations (based on reported peak sputum levels via nebulization in CF patients) were: colistin 40 µg/mL, levofloxacin 4,500 µg/mL, and tobramycin 1,000 µg/mL (31-33).”	Page 4. Lines 19-20.
Reviewer 2: The abstract is not formatted as should be for Microbiology Spectrum articles.	We thank the reviewer for this comment. The abstract has had the subheadings removed.	Page 1. Lines 16-33.
Reviewer 3: The use of only 14 isolates is a limitation, particularly given the authors' aim to compare GH activities "across diverse clinical isolates." A larger number of strains would strengthen the conclusions about differential GH activity against Psl and Pel, especially as some strains may produce minimal Pel EPS. It would also	We thank the reviewer for this insightful comment. We agree that a larger panel of clinical isolates would further strengthen conclusions about the relative activity of glycoside hydrolases against diverse P. aeruginosa biofilms. However, we note that the use of 14 clinical isolates represents a relatively large sample size for functional in vitro biofilm studies, especially compared to prior reports that typically use only a handful of laboratory and well-characterized clinical strains. Our goal was to capture a broader, more representative range of matrix phenotypes from early CF infections without pre-selecting isolates based on known exopolysaccharide production, thereby increasing the translational relevance of our findings.	Page 4. Lines 23-34. Page 5. Lines 1-2.

be valuable to quantify the EPS content in the tested biofilms.	We also agree that direct quantification of EPS content would provide valuable mechanistic insight. However, given the heterogeneity of biofilm composition and the limited scalability of EPS extraction and quantification, our study instead employed glycoside hydrolase susceptibility as a functional proxy for dominant matrix composition. We have added a sentence to the manuscript to clarify this rationale: “This approach was chosen to allow high-throughput screening across multiple clinical isolates. Direct chemical analysis of matrix polysaccharides was not performed due to its complexity and limited feasibility in translational or clinical applications.”	
Reviewer 3: The authors reference prior work on dual GH treatments; it would strengthen the paper to include at least a subset experiment testing combined Psl_{G_h} + PelA_h on a representative strain.	We agree with the author and thank them for this comment. We actually had done the confocal experiments with mixed GH treatments, and we have included these in Supplementary Figures 2 and 3. These experiments were limited to two strains and were therefore included in the Supplementary Materials rather than the main text, as we did not feel the findings could be broadly generalized across all isolates. We have included the following in our results section: “To assess whether combining glycoside hydrolases further enhanced biofilm disruption, we tested dual Psl_{G_h} + PelA_h treatments on two representative isolates (PA342 and PA380). As shown in Supplementary Figures 2 and 3, combined GH treatment did not result in additional biofilm reduction compared to the dominant single GH alone at equivalent concentrations.”	Page 12. Lines 14-18.
Reviewer 3: The mechanisms underlying the observed effects should be explored in greater depth. The interactions between antibiotics (e.g., colistin) and specific matrix components could be elaborated, as could the discussion of physicochemical alterations to the biofilm post-GH treatment (e.g., changes in porosity or charge repulsion).	We thank the reviewer for this comment. While we allude to some of these things in our discussion, we have now greatly expanded this. “Beyond charge effects, enzymatic degradation of the biofilm matrix may also alter biophysical properties that govern antibiotic penetration, including matrix porosity, density, and hydration. Prior work has shown that enzymatic disruption of biofilms can increase diffusivity by reducing matrix crosslinking and compactness (37-39). By cleaving specific polysaccharides (Psl or Pel), GHs may loosen the biofilm network, reduce steric hindrance, or modulate the binding affinity of matrix components for antibiotics. These physicochemical changes may partially account for the enhanced antibiotic susceptibility observed with GH pre-treatment. Although we did not measure matrix structure or permeability directly, our confocal imaging provides qualitative support for altered biofilm integrity following GH exposure. Future studies using high-resolution imaging, microviscosity measurements, or charge-mapping techniques could more precisely define these biophysical effects and help optimize GH-antibiotic pairings.”	Page 20. Lines 20-24. Page 21. Lines 1-6.
Reviewer 3: In Figure 2, consider including raw biomass values alongside percentage reductions for clearer	We thank the reviewer for this helpful suggestion. We agree that reporting only the relative percent reduction in biofilm biomass may obscure the variation in baseline biofilm-forming capacity between isolates. Our primary motivation for using relative	Page 9. Lines 1-4.

context.	reduction was to allow cross-strain comparisons in a standardized format, particularly given the large variability in absolute biofilm biomass and the challenge of graphically presenting all 14 isolates with different baseline OD ranges. To address the reviewer's concern, we have now included Supplementary Figure 1, which lists the average OD₆₀₀ values of untreated negative control wells for each of the 14 isolates. This provides transparency regarding baseline biofilm biomass and allows readers to contextualize the magnitude of biofilm reduction observed. We have also added a corresponding sentence in the Results section and updated the figure legends to point readers to Supplementary Figure 1. We have added the following sentence in the Results section: "To facilitate comparison across strains, biofilm biomass reduction is presented as a percentage relative to the untreated control for each isolate. However, as baseline biofilm formation varied between isolates, the average OD₆₀₀ values of the negative control wells for all 14 strains are provided in Supplementary Figure 1."	Supplementary Materials: Supplementary Figure 1.
Editor: Limited number of strains	We appreciate the reviewer's comment regarding the number of clinical isolates. While we acknowledge that 14 strains may appear modest, our study in fact expands the scope of prior work in this area. Most published studies evaluating GH activity have focused on laboratory strains or included only a few clinical isolates, often without systematic screening across early CF isolates. By contrast, our study offers a broader comparative assessment of GH activity across a diverse panel of early P. aeruginosa CF isolates, under conditions relevant to the CF airway environment.	
Editor: Concentrations of antibiotics	We thank the editor for this comment and believe that these concerns have been comprehensively addressed in the response to reviewer 2; however, if this is not the case, please let us know and we would be happy to provide further clarification. We have published previous work in AAC using these high concentrations of antibiotics seen in the CF airway (Tom, S.K., Yau, Y.C., Beaudoin, T., LiPuma, J.J. and Waters, V., 2016. Effect of high-dose antimicrobials on biofilm growth of Achromobacter species isolated from cystic fibrosis patients. Antimicrobial Agents and Chemotherapy, 60(1), pp.650-652). Our intention with these concentrations was to select concentrations that better reflect what is seen in vivo with these antibiotics, which are all used via nebulization in the CF clinic against P. aeruginosa.	Page 4. Lines 13-14.
Editor: The data presented are insufficient to define the strains as Pel- or Psl-dominant.	We appreciate the editor's thoughtful feedback and fully agree that our study does not provide direct biochemical evidence (e.g., via compositional analysis or exopolysaccharide quantification) to define the biofilm matrix composition of each isolate. Our use of the terms "Pel-dominant" or "Psl-dominant" was based on a functional definition, inferred from the degree of biofilm biomass reduction following treatment with PelA_n or PslG_n, respectively. Our goal was not to characterize EPS dominance at a molecular level, but rather to develop a practical screening approach that could inform targeted adjuvant therapy.	Throughout the manuscript.

	To clarify this, we have revised the manuscript to: 1. Use more cautious language around "dominance" (e.g., replacing "Pel-dominant" with or "PelA_n-disrupted").2. Emphasize in the Introduction and Discussion that this was a phenotypic screen rather than a mechanistic or compositional classification.3. Explicitly acknowledge the limitation that EPS dominance was inferred and not biochemically confirmed. We thank the editor again for this important clarification, which has improved the clarity and scope of our conclusions.	
--	---	--

Re: Spectrum01354-25R1 (**Glycoside Hydrolases Enhance Antibiotic Activity Against *Pseudomonas aeruginosa* Biofilms from Cystic Fibrosis Airways**)

Dear Dr. Isaac Martin:

I apologize for the delay. Based on the first reviews, your manuscript required a second round of evaluation. Unfortunately, securing reviewers in a timely manner proved challenging, as many were unavailable.

Below, you will find review comments along with instructions from the Spectrum editorial office. Please note that all critiques need to be addressed directly in the revised manuscript, not only in the rebuttal, as this is essential for the manuscript to be considered for publication. Please make sure to highlight all the edits in the marked version of the manuscript.

Revision Guidelines

Sincerely,
Marianna Patrauchan
Editor
Microbiology Spectrum

Reviewer #3 (Comments for the Author):

I have no additional comments

Reviewer #5 (Comments for the Author):

The manuscript by Isaac Martin et al. describes the impact of two glycoside hydrolases PslGh and PelAh on *P. aeruginosa* biofilms and evaluates their antibiofilm efficiency when applied in combination with colistin, levofloxacin, and tobramycin at their highest concentrations as achieved upon nebulization. The authors previously reported the combinatory application of the GH with antibiotics tested in acute murine model of infection, and this study tested 14 clinical strains isolated from pediatric patients. Overall, the focus of the study is of high clinical importance as enhancing the available antibiofilm approaches can be instrumental in reducing the bacterial burden during infection and improving prognosis.

However, some methodological details are alarming and need to be justified. (1) Clinical isolates were sub-cultured three times. This may cause occurrence of multiple mutations and make the described phenotypic features NOT reflecting the original isolates, as expression of PS is conditional. (2) Considering somewhat limited solubility of antibiotics, the preparation of such highly concentrated stocks needs to be described in the methods. (3) Syto-9 dye from ThermoFisher is a nuclear counterstain for bacterial Live-dead staining and stains both live and dead Gram-negative bacteria. It reads as the authors used Syto 9 alone without propidium iodide, which cannot report live cells only as claimed in the manuscript. Besides, the description of sample prep for imaging omits details of GH treatments, and the details on how "live-cell biovolumes" were quantified are missing. (4) CFU/ml data are presented, however no details on how cells were collected for this analysis were provided.

Western blotting was used to evaluate the stability of GHs in the presence of 10-20% of CF sputum supernatant. Although important, detection of the enzymes does not guarantee their activity, which is essential for evaluating their potential for clinical applications. Considering the scope of the paper, the authors are fully equipped to test the activity of the enzymes in the presence of CF sputum supernatant. This would strengthen the manuscript.

Another concern is that the manuscript contains a number of ambiguous statements or overstating conclusions, all of which need to be removed for scientific accuracy. Some examples are listed below and in "other comments"

The authors state that the aim was to assess the "relative contribution of Psl and Pel ... to biofilm formation" at least twice in the manuscript, however the provided results only show the impact of the corresponding GHs on biofilm volume, which is only suggestive and not direct evidence for Psl and Pel contributions.

The authors describe that 7 of the selected 14 isolates "went to cause chronic infection". What is the basis of such statement? Does fig 2 show the data for each strain as a dot? Then Table 1 shows the same data and is just calling the effect "additive" if the corresponding point in fig 2 shows difference more than 0. These changes are very modest, ranging from about 5 to about 20%. Although the table shows no inhibitory impact and the authors highlight this as a conclusion, the fig 2 clearly shows some dots in the negative, suggesting that the biofilm volume was increased and not decreased upon the enzymatic treatment. This removes confidence from most of the conclusions the authors made based on this analysis. Therefore, all the following statements about the "significant" effect of the enzymes sound groundless.

On P. 16 L.13 The CFU analysis only confirmed the microscopy-based observations for high dose levofloxacin but not others, as stated.

On P. 16 L. 15-16, the authors state that additive effects of PslGh-antibiotic combinations were only observed at higher GH levels. However, in Fig 3, there are several conditions where 0.1 dose had an effect similar to 0.5. Ex low-colistin, low-levofloxacin.

On P. 19 L11 "across all antibiotics" is not supported as according to fig 4, colistin is an exception.

On P. 19 L13, "this was supported by CFU reduction" is not supported for all the conditions, as no difference was observed at 0.1 high levofloxacin.

On P. 19 L. 16-17 the concentration-dependence does not seem to be supported, as in this case, the effect shall be proportional to the concentration. However, in the results, there is either effect at high concentration (at 0.5 high colistin and levofloxacin) or both concentrations show the same effect (0.1 levofloxacin) or there is no difference between concentrations (tobramycin).

On P. 20 L 19 The statement on partial degradation at 20 h is not supported by the results. There is partial degradation at 0 h with almost complete degradation at 20 h.

On P. 20 L 20 The statement on stability in LB is not supported as there is some degradation in LB at 4 and 20 h

P. 21 L 9 "Psl predominated" is not supported; L. 11 states "meaningful adjuvant activity" - does 10% difference make it meaningful? Which results "highlight its potential value"?

P. 21 L.13 This statement is much stronger than results. The mentioned enhancement, if occurred, was only by about 6-18%, with many strains showing no effect. The following discussion of physicochemical interactions is speculative. L. 18 Again, the statement on GH-colistin combination is not fully reflecting the data, as pelA had very modest effect with all three antibiotics. By the way, how could these data reflect the actual concentrations of colistin in CF?

P. 23 L. 8 The results would have implications for potential clinical use only after their activity in CF sputum is confirmed. L. 10, 12 I do not see any data supporting synergistic effect, and no data supporting the suggested administration of dual GH treatment. Besides, as stated in L. 22, PelAh "underwent significant degradation in the presence of only 10-20% of SF sputum supernatant", which negates clinical applicability of the enzyme.

Where is the data for the experiment addressing "a range of antibiotic concentrations" described on P. 13 L. 8-10?

Supplementary fig 2 and 3 seem to only cover dual treatment.

Using human samples requires ethical statement.

Other comments:

P 2 L. 30 and throughout. Need to clarify "early-stage clinical isolates" or "early" to avoid ambiguity.

P 3 L. 22-23 It would help to elaborate and clarify the "nanomolar activity in vitro and in vivo", as it is important in evaluating the

novelty and the impact of the presented results.

P4 L. 12 It is not clear why the selected 14 isolates would present "a broad range". This is a misleading description.

P 4 L 20 and throughout: Need to clarify "first time" clinical isolates to avoid misinterpretation.

P 7 L. 22 Protein stability cannot be quantified by densitometry as said, but detecting protein and evaluating their abundance can be used to assess stability. Rephrase for accuracy

P. 8 L. 13 Such generalized statements make room for mistakes. For accuracy, need to add "tested" The same applies to similar statement throughout the manuscript.

P. 8 L. 17 Need to include the concentration applied for the 1-h exposure

P. 10 L. 3 shall be "more tested clinical isolates"

P. 10 L. 4 shall be "mediated"

P 13 L13. This statement may imply that a multiple-dose screen was also performed.

P. 13 L. 19 It will help to report concentrations here.

P. 20 L. 2 "rapidly degrade" needs either data or reference. L. 3 "longer timeframe" needs data or reference. How does it compare to the "rapid" hours vs minutes?

P. 20 L 7 Either merge this paragraph with the above or explain the "conditions"?

Fig. 5 Would help to label figures with protein names

Manuscript
Number

Manuscript Title

Spectrum01354-25 Glycoside Hydrolases Enhance Antibiotic Activity Against *Pseudomonas aeruginosa* Biofilms from Cystic Fibrosis Airways

EDITOR/REVIEWER COMMENTS Each of the editor and the reviewer queries.	AUTHOR'S RESPONSE Response to the editor and reviewer queries here.	REFERENCE PAGE AND LINE Where the change appears in the revised manuscript.
Reviewer 5: The authors state that the aim was to assess the "relative contribution of Psl and Pel ... to biofilm formation" at least twice in the manuscript, however the provided results only show the impact of the corresponding GHs on biofilm volume, which is only suggestive and not direct evidence for Psl and Pel contributions.	We thank the reviewer for this important clarification. We agree that our results assess the functional response of biofilms to glycoside hydrolase (GH) treatment, rather than providing direct biochemical evidence of Psl or Pel content. Our aim was to infer the relative contribution of these matrix components based on the extent of biofilm disruption by PslG_h or PelA_h, recognizing that this represents an indirect, phenotypic measure. To make this distinction clearer, we have revised the manuscript to:  1. Use more cautious terminology throughout (e.g., replacing "Psl-dominant" or "Pel-dominant" with "PslG_h-disrupted" or "PelA_h-disrupted"). 2. Emphasize in the Introduction and Discussion that the study provides a functional screening approach rather than direct compositional evidence. 3. Explicitly acknowledge this limitation in the Discussion, noting that biochemical or imaging-based EPS quantification would be required to confirm matrix composition. We appreciate the reviewer's observation, which has helped refine the framing of our study and clarify its scope.	Throughout manuscript
Reviewer 5: P4 L. 12 It is not clear why the selected 14 isolates would present "a broad range". This is a misleading description.	We thank the reviewer for this comment. We used the term "broad" in a relative sense, to emphasize that our panel of 14 first-time P. aeruginosa clinical isolates represents a wider range of CF patient-derived strains than has been examined in prior GH-antibiofilm studies, which were largely limited to laboratory strains (e.g., PAO1, PA14) and a small number of clinical isolates. All isolates in our study were derived from distinct patients with first P. aeruginosa acquisition, and thus encompass a broad sampling of early infection phenotypes within this clinical context. We have added wording in the Methods to clarify this intended meaning. Fourteen first-time P. aeruginosa clinical isolates from distinct pediatric CF patients were selected to provide a broad representation of early infection phenotypes, extending beyond the limited number of clinical strains previously tested in GH-	Page 4 Lines 5-7

	antibiofilm studies.	
Reviewer 5: The authors describe that 7 of the selected 14 isolates "went on to cause chronic infection". What is the basis of such statement?	We thank the reviewer for requesting clarification. All 14 isolates were obtained from children with cystic fibrosis during their first P. aeruginosa infection episode. Each patient underwent standardized antibiotic eradication therapy per our institutional protocol (28 days of inhaled tobramycin or, if symptomatic, two weeks of dual intravenous antipseudomonal therapy followed by 28 days of inhaled tobramycin). "Chronic infection" versus "eradicated" status was determined through subsequent clinical follow-up and microbiological surveillance. In brief, seven patients failed eradication and converted to chronic colonization, whereas seven cleared the infection and remained culture-negative during follow-up. We have refined the sentence as follows: Fourteen first-time P. aeruginosa isolates from airway cultures of children with cystic fibrosis were selected - seven from patients who subsequently developed chronic infection despite standard eradication therapy and seven from patients in whom infection was successfully eradicated - as previously described (29, 30).	Page 4 Lines 5-8
Reviewer 5: Clinical isolates were sub-cultured three times. This may cause occurrence of multiple mutations and make the described phenotypic features NOT reflecting the original isolates, as expression of PS is conditional.	We thank the reviewer for this comment. We routinely revived each clinical isolate from the -80 °C master stock and performed three standardized passages before experiments. This is standard microbiological practice to (i) recover cells from cryostress, (ii) eliminate freeze-carryover antibiotics/cryoprotectants, and (iii) synchronize growth phase before phenotyping (Andrews et al. Antimicrobial susceptibility testing in practice. J Antimicrob Chemother. 2001). The total number of generations over these passages is modest; using published baseline mutation rates for P. aeruginosa (~10⁻⁹/bp/generation; genome ~6.3 Mb), the expected de novo mutations per lineage during this interval are <<1, making passage-induced genetic drift vanishingly unlikely to systemically alter phenotypes (Dettman et al., Genome-wide patterns of recombination in Pseudomonas aeruginosa. Genome Biol Evol. 2015). To address this response, we have added to the methods text: Following three subcultures as per standard microbiological practice (31), a single colony was inoculated into 4 mL lysogeny broth (LB; BioShop, Burlington, Canada) and incubated overnight at 37°C with shaking (225 rpm).	Page 4 Line 11
Reviewer 5: Considering somewhat limited solubility of antibiotics, the preparation of such highly concentrated stocks needs to be described in the methods.	We thank the reviewer for this comment. We have included some more methodology in the methods section: Stock solutions were prepared per manufacturer instructions. Tobramycin sulfate was dissolved in sterile water at 50 mg/mL, filter-sterilized (0.22 µm), aliquoted, and stored at -20 °C. Colistin sulfate was dissolved in sterile water at 10 mg/mL and	Page 5 Lines 6-12

	stored at 4 °C, prepared the week of the experiments. Levofloxacin was prepared at 50 mg/mL in sterile water with mild NaOH adjustment to facilitate dissolution (final pH 6.8–7.2), then filter-sterilized and stored at –20 °C protected from light. Initial working concentrations used in assays were chosen to reflect upper-airway levels reported after nebulization (colistin 40 µg/mL, levofloxacin 4,500 µg/mL, and tobramycin 1,000 µg/mL) (33-35).	
Besides, the description of sample prep for imaging omits details of GH treatments, and the details on how "live-cell biovolumes" were quantified are missing.	We thank the reviewer for this comment. Details of GH treatments are included in the methods section: Hydrolases were also tested at low and high concentrations of 0.1 and 0.5 mg/mL, respectively, to evaluate whether higher enzyme levels produced greater biofilm disruption. We agree that more details in reference to the quantification of live cell biovolumes would benefit from further clarification. The section now reads: The integrated SYTO 9-positive voxel volume (µm³) was recorded as "live-cell biovolume," reflecting the biomass of membrane-intact cells visualized in the green channel. This metric was supported by corresponding CFU assays, which demonstrated a strong positive correlation (Pearson $r \approx 0.9$) between CFU/mL and SYTO 9-derived biovolume across treatment conditions.	Page 7 Lines 12-16
Reviewer 5: Syto-9 dye from ThermoFisher is a nuclear counterstain for bacterial Live-dead staining and stains both live and dead Gram-negative bacteria.	Response: We apologize that this was unclear. We indeed used the LIVE/DEAD BacLight kit (SYTO 9 + propidium iodide) and quantified "live-cell biovolume" from the SYTO 9 channel after adjusting for PI. We have revised the Methods and figure legends to state the two-dye protocol explicitly. Ammended Methods detail: Confocal imaging and quantification of live-cell biovolume After 18 h incubation at 37°C, wells were washed gently and biofilms were stained with the LIVE/DEAD BacLight bacterial viability kit (SYTO 9 and propidium iodide; Thermo Fisher, Mississauga, ON) in LB for 45 min at room temperature. Wells were then gently washed and replaced with 200 µL of fresh LB for imaging. Both channels were acquired using identical laser and gain settings on a Quorum confocal laser scanning microscope (Zeiss AxioVert 200 M, 25× objective; total magnification ×250; 0.3 µm z-steps). The PI channel was used during acquisition to visually identify and adjust for PI-positive (non-viable) material, and quantification was then performed from the SYTO 9 channel under fixed imaging parameters across all conditions. Image stacks were processed in Velocity software.	Page 7 Lines 4-5 Lines 9-11

Reviewer 5: CFU/ml data are presented, however no details on how cells were collected for this analysis were provided.	We thank the reviewer for this comment. We have added the following sentence to the methods section, referencing the seminal work by O’Toole in 2011: Corresponding CFU assays were performed on parallel biofilms grown on glass chamber slides under identical conditions. Following treatment, biofilms were scraped into PBS, serially diluted, and plated on LB agar for colony enumeration using standard methods (36).	Page 7 Lines 13-16
Reviewer 5: Although the table shows no inhibitory impact and the authors highlight this as a conclusion, the fig 2 clearly shows some dots in the negative, suggesting that the biofilm volume was increased and not decreased upon the enzymatic treatment.	We appreciate the reviewer’s careful examination of Figure 2. For this figure, many of the GH treatments in which there was no treatment effect do indeed cluster around zero. The negative points indeed reflect the natural variability between biological replicates; however, none of these differences reached statistical significance relative to untreated controls (two-tailed test, $p > 0.05$). We have clarified this in the results text to avoid any misinterpretation that enzymatic treatment increased biofilm formation in a significant manner. We have now explicitly stated in the Results that “no statistically significant increase in biofilm biomass was observed under any condition.” We would also like to draw the reviewer’s attention to Supplementary Figure 1 (with raw OD readings), which indicates exactly which experiments reached statistical significance and what the degree of significance was.	Page 10 Lines 2-5
Reviewer 5: On P. 19 L11 "across all antibiotics" is not supported as according to fig 4, colistin is an exception.	Many thanks to the reviewer for highlighting this potential ambiguity. Using the two assays, there were examples of additivity across all antibiotics. There were examples of biofilm reduction for levofloxacin and tobramycin using confocal microscopy and for colistin and levofloxacin using CFU quantification. This has been clarified in the main text by the addition of “across both assays”.	Page 19 Line 11
Reviewer 5: Western blotting was used to evaluate the stability of GHs in the presence of 10-20% of CF sputum supernatant. Although important, detection of the enzymes does not guarantee their activity, which is essential for evaluating their potential for clinical applications. Considering the scope of the paper, the authors are fully equipped to test the activity of the	We thank the reviewer for this valuable suggestion. We agree that enzyme detection by Western blot does not directly confirm retained activity. To address this, we did perform an additional 1-hour biofilm disruption assay using the same CF sputum supernatant concentrations (10–20%) and methodology as for the Western blot experiments. These results have been included in the Supplementary Materials (Supplementary Table 4) and referenced in the Results and Discussion sections.	Supplementary materials Table 3 Page 21 Lines 3-11

enzymes in the presence of CF sputum supernatant. This would strengthen the manuscript.		
Reviewer 5: On P. 19 L13, "this was supported by CFU reduction" is not supported for all the conditions, as no difference was observed at 0.1 high levofloxacin.	Many thanks to the reviewer for this level of scrutiny. Broadly speaking, our findings for biovolume were supported by CFU reduction.	
Reviewer 5: On P. 19 L. 16-17 the concentration-dependence does not seem to be supported, as in this case, the effect shall be proportional to the concentration. However, in the results, there is either effect at high concentration (at 0.5 high colistin and levofloxacin) or both concentrations show the same effect (0.1 levofloxacin) or there is no difference between concentrations (tobramycin).	We thank the reviewer for this comment. The sentence has now been clarified. There was indeed no definitive concentration dependence seen. We would also draw the reviewer's attention to the discussion: A clear dose-response relationship was not consistently observed, raising questions about biofilm accessibility or whether target saturation occurs at lower GH concentrations.	Page 19 Lines 16-18
Reviewer 5: On P. 20 L 19 The statement on partial degradation at 20 h is not supported by the results. There is partial degradation at 0 h with almost complete degradation at 20 h.	We thank the reviewer for this comment. We feel that we are justified in making this statement for two reasons: (1) in our disruption assay, full biofilm degrading properties of this GH was seen at the 20 hour time point; and (2) there is still a visible band at 20 hours.	
Reviewer 5: On P. 20 L 20 The statement on stability in LB is not supported as there is some degradation in LB at 4 and 20 h	We thank the reviewer for this comment. There is indeed some degradation of the band in LB; however, we saw that the biofilm degrading properties of this GH was maintained at both the 4 and 20 hour time points in our functional assay.	
Reviewer 5: P. 21 L 9 "Psl predominated" is not supported;	We thank the reviewer for this comment. The sentence has now been changed to: PslG _h -mediated disruption predominated.	Page 21 Line 9
Reviewer 5: P21 L. 11 states PelA _h had "meaningful adjuvant activity" - does 10% difference make it meaningful? Which results "highlight its potential value"?	We appreciate the reviewer's request for clarification. Although PelA _h reduced biofilm biomass significantly in only 3 of the 14 clinical isolates, these effects were statistically significant and biologically meaningful within those strains, corresponding to enhanced antibiotic susceptibility when combined with tobramycin and levofloxacin. Importantly, these PelA _h -responsive isolates were distinct from those disrupted by PslG _h , indicating that PelA _h targets a complementary subset of biofilm phenotypes. In the representative "PelA _h -disrupted" strain (PA380), confocal imaging and CFU assays further confirmed instances of additive biofilm disruption across the	Results and Discussion

	antibiotics tested. Thus, our description of PelA_h as having “meaningful adjuvant activity” refers to its strain-specific but reproducible enhancement of antibiotic efficacy, highlighting its potential value as part of a dual-GH therapeutic approach, rather than implying uniform potency across all isolates. We have clarified this nuance in the Discussion section.	
Reviewer 5: P. 21 L.13 This statement is much stronger than results. The mentioned enhancement, if occurred, was only by about 6-18%, with many strains showing no effect. The following discussion of physicochemical interactions is speculative. L.	We thank the reviewer for this comment and have removed the word “significantly”, although we did see statistically significant differences. The following discussion of physicochemical interactions is speculative and that’s why it is limited to the discussion section, where such speculations can be discussed.	Page 21 Line 19
Reviewer 5: By the way, how could these data reflect the actual concentrations of colistin in CF?	We thank the reviewer for this comment. Although there are many references cited in the article that look at the concentrations of these antibiotics in the airways of patients with CF, we would recommend the article Chmiel et al. (Annals ATS, 2014), which does a good job of summarizing the existing data. We have also discussed colistin specifically in the discussion and why this antibiotic has a much lower peak sputum concentration than both tobramycin and levofloxacin: This may reflect reduced colistin concentrations in airway secretions due to its large molecular size and poor sputum bioavailability compared to levofloxacin or tobramycin	
Reviewer 5: Using human samples requires ethical statement.	We thank the reviewer for this comment. Please refer to the REB number provided in the methods section. The bacterial isolates contain no human biological material or identifiable information and are not considered “human samples” under our institutional or national research ethics frameworks. As such, additional human-subject ethical statements or consent procedures are not required.	
Reviewer 5: P 3 L. 22-23 It would help to elaborate and clarify the "nanomolar activity in vitro and in vivo", as it is important in evaluating the novelty and the impact of the presented results.	We thank the reviewer for this comment. We did not think that a discussion about GH concentrations was appropriate for the introduction section where the concentrations in our experiments are not mentioned; however, we did previously include in the methodology section that the concentrations used in these experiments are based on “concentrations used in previous studies disrupting biofilms to enhance antibiotic efficacy in vitro and in vivo (25-27).” We have further clarified this in the Discussion section: Our dosing strategy (0.05–0.1 mg/mL) mirrors concentrations shown to be effective	Page 21 Lines 17-19

	in vitro (25) and in murine infection models (26, 27), supporting the translational relevance of the GH activities observed here.	
Reviewer 5: Besides, as stated in L. 22, PelAh "underwent significant degradation in the presence of only 10-20% of SF sputum supernatant", which negates clinical applicability of the enzyme.	We thank the reviewer for raising this important point. We agree that PelA _h 's reduced stability in sputum supernatant warrants caution, and we have already addressed this in the Discussion section (where we talk about testing orthologues from other bacterial species that may be more stable). However, instability in airway secretions does not intrinsically preclude therapeutic applicability; several approved inhaled biologics require formulation strategies to maintain activity in mucus-rich environments. Our interpretation is therefore that PelA _h 's degradation highlights a key translational challenge rather than negating its potential use.	

Re: Spectrum01354-25R2 (Glycoside Hydrolases Enhance Antibiotic Activity Against *Pseudomonas aeruginosa* Biofilms from Cystic Fibrosis Airways)

Dear Dr. Isaac Martin:

Thank you for the privilege of reviewing your work. Below you will find my comments and instructions from the Spectrum editorial office.

I have reviewed the revised manuscript. While many of the reviewers' concerns have been addressed, several issues remain unresolved. In particular, the manuscript continues to include instances of vague or inflated language. Accordingly, the manuscript cannot be accepted for publication at this time.

As Microbiology Spectrum is an ASM journal, we have a responsibility to uphold high standards of scientific rigor and clarity. Should you choose to continue with Microbiology Spectrum, all the issues outlined below need to be fully addressed. Please clearly highlight all revisions in the manuscript and provide a point-by-point response to each concern, following the order in which they are listed below.

First, I did not find the following comments from Reviewer 5 addressed in the authors' response table, and because not all revisions were tracked, it was not easy to determine if they were addressed. Please ensure that all these comments are addressed or explain why they were not addressed. I am listing them as they were with the page/line numbers of the previous version.

On P. 16 L.13 The CFU analysis only confirmed the microscopy-based observations for high dose levofloxacin but not others, as stated.

On P. 16 L. 15-16, the authors state that additive effects of PslGh-antibiotic combinations were only observed at higher GH levels. However, in Fig 3, there are several conditions where 0.1 dose had an effect similar to 0.5. Ex low-colistin, low-levofloxacin.

L. 18 Again, the statement on GH-colistin combination is not fully reflecting the data, as pelA had very modest effect with all three antibiotics.

L. 10, 12 I do not see any data supporting synergistic effect, and no data supporting the suggested administration of dual GH treatment.

Where is the data for the experiment addressing "a range of antibiotic concentrations" described on P. 13 L. 8-10? Supplementary fig 2 and 3 seem to only cover dual treatment.

P 4 L 20 and throughout: Need to clarify "first time" clinical isolates to avoid misinterpretation.

P 7 L. 22 Protein stability cannot be quantified by densitometry as said, but detecting protein and evaluating their abundance can be used to assess stability. Rephrase for accuracy

P. 8 L. 13 Such generalized statements make room for mistakes. For accuracy, need to add "tested" The same applies to similar statement throughout the manuscript.

P. 8 L. 17 Need to include the concentration applied for the 1-h exposure

P 13 L13. This statement may imply that a multiple-dose screen was also performed.

P. 13 L. 19 It will help to report concentrations here.

P. 20 L. 2 "rapidly degrade" needs either data or reference. L. 3 "longer timeframe" needs data or reference. How does it compare to the "rapid" hours vs minutes?

Second, several reviewer's comments were only partially addressed. This includes the comment on Psl and Pel contributions to biofilm formation. There are at least three instances, where it is still mentioned as the goal of this study: P. 3 L.18-19 states "to assess the relative contribution of Psl and Pel PS to biofilm formation"; p.8 L.19; p.21 L18-19.

The point on testing a broader range of clinical isolates than in other studies requires citations of those studies (p. 4 L.7).

The results need to be described more accurately and not with "broad strokes". For example, the sentence on P.19 L.11-12 states a significant biofilm reduction with PelAh across both assays and across all antibiotics. However, as the reviewer highlighted, this is not true for colistin (both biovolume and CFU at low concentration). In addition, more clarity is needed to relate/distinguish statistically vs biologically significant results here and throughout the manuscript.

In fig 5, I see evidence of partial degradation of PslGh at 0h when incubated with 10 and 20% of sputum (multiple bands) and significant degradation at 20 h (faint band). The observation of PslGh effect on biofilm after 20 h exposure to CF-sputum cannot disprove these data but rather suggests that the remaining amount of the enzyme may have an impact. Further, as shown in Supp table 3, PslGh impact on biofilm at 20h after exposure to 10 and 20% sputum is ~11{plus minus}4 and 11{plus minus}8 %, which, considering the recognized low accuracy of biofilm assay, is not statistically/biologically significant. The same is apparent for PelAh: none of the impact at 0h seem significant and shall not be shown in green. Further, it is not clear how "full" and "partial" enzymatic activities were defined. This table also needs to show statistical analysis.

The authors argue that the reported ~6-18% reduction in biofilm volume supports conclusions regarding meaningful adjuvant activity and therapeutic potential. However, considering the limited accuracy of the biofilm quantification approaches employed, changes of this magnitude (particularly near ~10%) may fall within the range of experimental errors. Accordingly, these interpretations appear overstated and need to be adjusted to align more closely with the data.

Finally, I would like to request the following revisions.

It is unclear whether Fig. 2 reports % of additional or combined biomass reduction achieved by enzyme treatment in the presence of antibiotics? As written, the text suggests a combined effect, but this needs to be explicitly clarified. Pls provide a description of how exactly the values in Figure 2 were calculated. It is also unclear how a paired t-test could be appropriately applied if antibiotic-alone controls were not included here. Pls clarify.

P.6 L 24 Pls describe the laser/gain settings.

P. 8 L13 Pls provide the name of the center.

P.8 L. 21 I would delete "adherence", as the assay was used to evaluate biofilm formation.

The Results section would benefit from more explicit references to specific figures and figure panels, as the current presentation makes it difficult to directly relate the text to the data.

LB is known to provide background fluorescence, and it is unclear why it was added to samples prior to confocal microscopy. This requires clarification and justification.

When biofilms were scraped for CFU, it is not stated whether the efficiency of this procedure was controlled. Please clarify or acknowledge potential error.

Referring to strains as "diverse" requires supporting sequence-based analysis.

P. 12 L. 5 It is unclear how the t-test was applied to "examine the results" shown in table 1.

P. 13 L. 12 The wording "most functionally implicated" needs rephrasing for accuracy.

P 22 L.6-7 Why was negatively charged alginate, commonly produced by *P. aeruginosa* clinical isolates, not considered in this discussion?

Revision Guidelines

Sincerely,

Marianna Patrauchan
Editor
Microbiology Spectrum

**Manuscript
Number**

Manuscript Title

Spectrum01354-25 Glycoside Hydrolases Enhance Antibiotic Activity Against Pseudomonas aeruginosa Biofilms from Cystic Fibrosis Airways

EDITOR/REVIEWER COMMENTS Each of the editor and the reviewer queries.	AUTHOR'S RESPONSE Response to the editor and reviewer queries here.	REFERENCE PAGE AND LINE Where the change appears in the revised manuscript.
On P. 16 L.13 The CFU analysis only confirmed the microscopy-based observations for high dose levofloxacin but not others, as stated.	We apologise that some of our previous track changes did not come through. The sentence currently reads: "CFU analysis confirmed reductions in viable cell counts seen with levofloxacin (500 µg/mL) when co-administered with both the low and high PslGh concentrations."	Page 17, Line 12-14
On P. 16 L. 15-16, the authors state that additive effects of PslGh-antibiotic combinations were only observed at higher GH levels. However, in Fig 3, there are several conditions where 0.1 dose had an effect similar to 0.5. Ex low-colistin, low- levofloxacin.	We apologise for the same issue here. The sentence currently reads: "Significant reductions in live cell biovolume were observed for all three antibiotics when co-administered with PslGh under specific concentration conditions."	Page 17, Line 11-13
L. 18 Again, the statement on GH-colistin combination is not fully reflecting the data, as pelA had very modest effect with all three antibiotics.	We thank the reviewer for this comment. What we have done is changed the introductory sentence to be more specific: "PelAh, but not PslGh, was associated with statistically significant reductions in biofilm biovolume and/or viable counts in combination with levofloxacin, tobramycin, and colistin under specific concentration conditions."	Page 20, Lines 11-13
L. 10, 12 I do not see any data supporting synergistic effect, and no data supporting the suggested administration of dual GH treatment.	Our experiments were not designed to detect synergy. We could not find reference to synergistic effects in this document. We did perform dual GH treatment and the sentence that describes this is in the results section: "We also tested dual PslGh + PelAh treatments on both representative isolates (PA342 and PA380). As shown in Supplementary Figures 2 and 3, combined GH treatment did not result in additional biofilm reduction compared to the single GH alone at equivalent concentrations."	Page 14, Line 17-20
Where is the data for the experiment addressing "a range of antibiotic	We apologise for the track changes issue.	Page 14, Line 10-12

concentrations" described on P. 13 L. 8-10? Supplementary fig 2 and 3 seem to only cover dual treatment.	The sentence currently reads: "We also tested two antibiotic concentrations to reflect the variable drug levels that may occur in different lung regions following inhalation therapy."	
P 4 L 20 and throughout: Need to clarify "first time" clinical isolates to avoid misinterpretation.	We apologise for the track changes issue. The sentence currently reads: "Fourteen first-time P. aeruginosa clinical isolates from distinct pediatric CF patients were selected to provide a broad representation of early infection phenotypes, extending beyond the limited number of clinical strains previously tested in GH-antibiofilm studies (21, 23, 25-30)."	Page 4, Line 6-8
P 7 L. 22 Protein stability cannot be quantified by densitometry as said, but detecting protein and evaluating their abundance can be used to assess stability. Rephrase for accuracy	We have changed the title of this section. The title of this section is now: "Determining the protein integrity of PslGh and PelAh when incubated with CF sputum supernatant"	Page 8, Line 3
P. 8 L. 13 Such generalized statements make room for mistakes. For accuracy, need to add "tested" The same applies to similar statement throughout the manuscript.	We apologise for the track changes issue. The sentence currently reads: "We screened 14 early P. aeruginosa clinical isolates from individuals with CF using functional enzymatic disruption assays to infer strain-specific dependence on the Psl and Pel exopolysaccharide in mature biofilms."	Page 9, Line 12-14
P. 8 L. 17 Need to include the concentration applied for the 1-h exposure	We apologise for the track changes issue. The sentence currently reads: "Recombinant PslGh and PelAh at a concentration of 0.1 mg/mL were incubated in PBS, or in PBS with 10% or 20% sputum supernatant, at 37°C. Aliquots were collected at 0, 4, and 20 h."	Page 8, Line 6-8
P 13 L13. This statement may imply that a multiple-dose screen was also performed.	We apologise for the track changes issue. The sentence currently reads: "We also tested two antibiotic concentrations to reflect the variable drug levels that may occur in different lung regions following inhalation therapy."	Page 14, Line 8-10
P. 13 L. 19 It will help to report concentrations here.	We thank the reviewer for this comment. We have now added the following: "For PA342, low/high antibiotic concentrations (µg/mL) were colistin 8/40, levofloxacin 100/500, and tobramycin 100/500; for PA380, concentrations were colistin 8/40, levofloxacin 100/500, and tobramycin 200/1000."	Page 14, Line 10-12

P. 20 L. 2 "rapidly degrade" needs either data or reference. L. 3 "longer timeframe" needs data or reference. How does it compare to the "rapid" hours vs minutes?	We apologise once again for the track changes issue. The sentence currently reads: "Although both PslGh and PelAh rapidly degrade their respective matrix exopolysaccharides (5-15 minutes) (25-27), antibiotic action occurs over a longer timeframe, particularly in biofilms (hours to days) (39)."	Page 21, Line 2-4
There are at least three instances, where it is still mentioned as the goal of this study: P. 3 L.18-19 states "to assess the relative contribution of Psl and Pel PS to biofilm formation"; p.8 L.19; p.21 L18-19.	We thank the editor for this comment. How these sentences read now: In the Introduction: "In this study, we used functional disruption assays to compare the relative functional dependence of early P. aeruginosa CF isolates on Psl versus Pel on biofilm formation, and to evaluate whether enzymatic disruption of these matrix components could enhance antibiotic activity." In the Results: "We screened 14 early P. aeruginosa clinical isolates from individuals with CF using functional enzymatic disruption assays to infer strain-specific dependence on the Psl and Pel exopolysaccharide in mature biofilms." And in the Discussion: "In our functional assay, we demonstrate that early P. aeruginosa CF isolates exhibit functional dependence on both Psl and Pel, with PslGh-mediated disruption predominating."	Page 3, line 18 Page 9, Line 9-11 Page 22, Line 18-20
The point on testing a broader range of clinical isolates than in other studies requires citations of those studies (p. 4 L.7).	We thank the editor for pointing this out. References added (21, 23, 25-30).	Page 4, Line 8
The sentence on P.19 L.11-12 states a significant biofilm reduction with PelAh across both assays and across all antibiotics. However, as the reviewer highlighted, this is not true for colistin (both biovolume and CFU at low concentration). In addition, more clarity is needed to relate/distinguish statistically vs biologically significant results here and throughout the manuscript.	We thank the editor for this comment. Regarding statistical versus biological significance, we wish to clarify that our Results are limited to reporting statistically significant effects based on predefined analyses. We do not infer biological or clinical significance from these in vitro findings. To avoid ambiguity, we have confined discussion of potential translational relevance to the Discussion, where it is framed cautiously and hypothesis-generating. "PelAh, but not PslGh, was associated with statistically significant reductions in biofilm biovolume and/or viable counts in combination with levofloxacin, tobramycin, and colistin under specific concentration conditions."	Page 20, Line 11-13

In fig 5, I see evidence of partial degradation of PslGh at 0h when incubated with 10 and 20% of sputum (multiple bands) and significant degradation at 20 h (faint band). The observation of PslGh effect on biofilm after 20 h exposure to CF-sputum cannot disprove these data but rather suggests that the remaining amount of the enzyme may have an impact. Further, as shown in Supp table 3, PslGh impact on biofilm at 20h after exposure to 10 and 20% sputum is $\sim 11 \pm 4$ and 11 ± 8 %, which, considering the recognized low accuracy of biofilm assay, is not statistically/biologically significant. The same is apparent for PelAh: none of the impact at 0h seem significant and shall not be shown in green. Further, it is not clear how "full" and "partial" enzymatic activities were defined. This table also needs to show statistical analysis.	We thank the editor for this thoughtful comment. The purpose of sputum supernatant incubation experiments was to assess whether CF sputum components promote degradation of PslGh and PelAh, and whether residual enzymatic activity could still be detected following exposure. These assays were not designed to quantify absolute enzymatic activity or establish equivalence between protein abundance and functional effect. We agree that effects in this supplementary confirmatory assay for PelAh are modest and should be interpreted cautiously. However, we note that statistical "significance" is not synonymous with effect magnitude, and modest ($\sim 10\%$) differences can be statistically significant depending on variance and pairing. We intentionally did not perform formal statistical testing for Supplementary Table 3 because these assays were designed as ancillary, qualitative confirmation of residual activity after sputum exposure and involved multiple conditions (time point, sputum concentration, enzyme), where multiple hypothesis testing could encourage over-interpretation (and inflation of type I error). Accordingly, we present these data descriptively (mean \pm SEM), have removed color coding and descriptions of activity retained, and have clarified in the table legend and text that the purpose is to illustrate trends in residual activity rather than to infer biological/clinical significance. We have changed the title of the section to: "Differential susceptibility of PslGh and PelAh to degradation by CF sputum supernatant" And the body of the text now reads: "PslGh remained detectable at all time points in LB alone, with almost full degradation observed at 20 hours when incubated with CF sputum supernatant. In contrast, while PelAh was stable in LB with some degradation seen at the 4 and 20 hour time points, no detectable PelAh bands remained at 4 or 20 hours in the presence of either 10% or 20% sputum supernatant. These results indicate that while both enzymes are susceptible to degradation in CF airway conditions, PslGh demonstrates greater stability than PelAh, supporting its potential utility as a more robust therapeutic adjunct. To determine whether residual enzymatic activity could still be detected following exposure to CF sputum supernatant under the same conditions used for immunoblotting, we performed a functional biofilm disruption assay using laboratory strains with well-defined matrix architectures (PAO1 for PslGh and PA14 for PelAh). Following pre-incubation, PslGh-treated biofilms exhibited measurable disruption at early time points (0 and 4 hours) in both buffer and sputum-containing conditions, with reduced effects observed only after prolonged incubation (20 h), including in the	Supplementary Materials Table 3 Page 21, Line 2-33 Page 22, Line 1-14
---	---	---

	absence of sputum supernatant. In contrast, PelAh-mediated effects were diminished in sputum-containing conditions and were not consistently detectable at later time points. Results are presented descriptively in Supplementary Table 3.”	
The authors argue that the reported ~6-18% reduction in biofilm volume supports conclusions regarding meaningful adjuvant activity and therapeutic potential. However, considering the limited accuracy of the biofilm quantification approaches employed, changes of this magnitude (particularly near ~10%) may fall within the range of experimental errors. Accordingly, these interpretations appear overstated and need to be adjusted to align more closely with the data.	We thank the editor for this thoughtful comment. We agree that modest absolute reductions in biofilm volume should be interpreted cautiously, particularly given inherent variability in biofilm quantification assays. Our conclusions are based on replicated experiments demonstrating statistically significant and consistent reductions across conditions, and were not intended to imply standalone therapeutic efficacy. We have revised the text to emphasize that these effects support adjuvant activity in a proof-of-principle context, rather than direct clinical impact, and have moderated language regarding therapeutic potential accordingly. We have added the following in the discussion section: “Considering the limited accuracy of the biofilm quantification approaches employed, some of the significant changes observed may fall within the range of experimental errors. Also, given the inherent variability of biofilm quantification assays, effect sizes were interpreted in conjunction with statistical significance, antibiotic killing assays, and strain-specific structural biofilm disruption observed by confocal microscopy.”	Page 23, Line 2-7
It is unclear whether Fig. 2 reports % of additional or combined biomass reduction achieved by enzyme treatment in the presence of antibiotics? As written, the text suggests a combined effect, but this needs to be explicitly clarified. Pls provide a description of how exactly the values in Figure 2 were calculated. It is also unclear how a paired t-test could be appropriately applied if antibiotic-alone controls were not included here. Pls clarify.	We apologise if the Figure 2 did not display properly. We have taken the time to ensure that it is now accurate and has downloaded well in the current Figures we have uploaded. We do believe that this is now clear in the wording of both the text and in the figure itself. The Y axis title of Figure 2 is: “Percent (%) additional biomass reduction compared to antibiotic alone” The title of Figure 2 is: “The adjunctive benefit of co-administration of antibiotics with PelAh (white bars) and PslGh (grey bars) assessed by the crystal violet assay when compared to antibiotic alone.” Also in the figure legend, we say: “Paired t-tests were used to compare antibiotic + enzyme vs. antibiotic alone. *p<0.05; ***p<0.001”. In the body of the text, it reads: “To quantify the incremental effect of glycoside hydrolase co-administration, biofilm biomass reduction was calculated relative to antibiotic-alone controls under matched conditions. Using this approach, PslGh co-administration resulted in an additional 7.6	

	$\pm 10.0\%$ (95% CI), $15.7 \pm 9.2\%$ (95% CI; $p < 0.05$) and an $18.1 \pm 5.7\%$ (95% CI; $p < 0.001$) reduction in biofilm biomass when combined with colistin, levofloxacin and tobramycin, respectively. PelAh co-administration produced additional reductions of $6.0 \pm 5.3\%$ (95% CI), $7.7 \pm 3.8\%$ (95% CI) and $4.1 \pm 4.0\%$ (95% CI) when combined with colistin, levofloxacin and tobramycin, respectively.” We would be happy to provide any further clarification, but have represented the data in this way because we felt that the message is clearer when presented in this way.	
P.6 L 24 Pls describe the laser/gain settings.	We have now clarified the confocal acquisition parameters, including laser excitation and emission windows, and have specified that detector gain settings were held constant across all conditions. “Excitation was set at 488 nm (SYTO9) and 561 nm (propidium iodide), with emission collected at 500–540 nm and 600–650 nm, respectively. Laser power and detector gain settings were kept constant across all experimental conditions.”	Page 7, Line 13-15
P. 8 L13 Pls provide the name of the center.	The Hospital for Sick Children	Page 8, Line 21
P.8 L. 21 I would delete "adherence", as the assay was used to evaluate biofilm formation.	Replaced with “biofilm”	Page 10, Line 1
The Results section would benefit from more explicit references to specific figures and figure panels, as the current presentation makes it difficult to directly relate the text to the data.	We thank the editor for this comment and agree that explicit references would enhance the reading experience and allow for more ease in relating the data to the appropriate figure. We have now changed this throughout for Figures 3 and 4.	Page 17, Line 11-17 Page 20, Line 11-19
LB is known to provide background fluorescence, and it is unclear why it was added to samples prior to confocal microscopy. This requires clarification and justification.	We thank the editor for raising this point. LB was included during confocal imaging to maintain physiological growth conditions and avoid biofilm perturbation associated with media exchange or nutrient deprivation prior to imaging. Our confocal analyses were focused on biofilm structure and matrix disruption rather than metabolic activity, and all experimental conditions (including untreated controls) were imaged in the same medium. Any background fluorescence attributable to LB was adjusted for in image acquisition and consistent across conditions, thereby making it less likely to affect relative comparisons. We have now clarified this rationale in the Methods: “LB was selected over buffer-based media to avoid biofilm collapse or matrix rearrangement associated with nutrient withdrawal prior to imaging.”	Page 7, Line 7-8
When biofilms were scraped for CFU, it	We thank the editor for this comment and agree that this is a source of potential	Page 7, Line 18-20

is not stated whether the efficiency of this procedure was controlled. Please clarify or acknowledge potential error.	error. CFU enumeration was performed using widely used and well established mechanical disruption and plating methods already cited in this paper (O’Toole et al., J Vis Exp, 2011). We also performed these experiments in biological triplicate for rigour. Although complete recovery efficiency is not directly quantified, the same scraping and resuspension protocol was applied consistently across all conditions, enabling relative comparisons. We have clarified this point in the Methods and acknowledged this inherent limitation: “While complete recovery of all biofilm-associated cells cannot be assumed, the same procedure was applied uniformly across all conditions, allowing valid relative comparisons of viable bacterial burden.”	
Referring to strains as "diverse" requires supporting sequence-based analysis.	Have replaced the word “diverse” and have replaced with “genetically distinct”. We have also now cited the BioProject numbers from the WGS data in the Methods section. “All whole genome sequenced data from these isolates are available through Bioproject PRJNA556419 with the following Genbank accessions: JAGHM W000000000 (Pa50), JAGHMOV000000000 (Pa263), JAGHMU000000000 (Pa288), JAGHMT000000000 (Pa325), JAGHMS000000000 (Pa342), JAGHM R000000000 (Pa375), JAGHMQ000000000 (Pa380), JAGHMP000000000 (Pa404), JAGHMO000000000 (Pa505), JAGHMN000000000 (Pa551), JAGHM M000000000 (Pa558), JAGHML000000000 (Pa565), JAGHMK000000000 (Pa580), JAGHMI000000000 (Pa549).”	Page 3, Line 3 Page 22, Line 15 Page 4, Line 9-15
P. 12 L. 5 It is unclear how the t-test was applied to" examine the results" shown in table 1.	We thank the editor for pointing out this potential ambiguity. How the sentence now reads: “We next used paired T-tests to compare the effect of antibiotic + GH combination compared to antibiotic alone for each individual strain (Table 1).”	Page 13, Line 6
P. 13 L. 12 The wording "most functionally implicated" needs rephrasing for accuracy.	Changed to: “exopolysaccharide to which each strain showed the greatest susceptibility in the initial screen”.	Page 14, Line 13-14
P 22 L.6-7 Why was negatively charged alginate, commonly produced by P. aeruginosa clinical isolates, not considered in this discussion?	We thank the editor for this comment. This is a great point and has now been included in the discussion. “Alginate, a third major P. aeruginosa exopolysaccharide that predominates in mucoid phenotypes, is also highly anionic due to its uronic acid backbone (41).”	Page 23, Line 3-4

Re: Spectrum01354-25R3 (**Glycoside Hydrolases Enhance Antibiotic Activity Against *Pseudomonas aeruginosa* Biofilms from Cystic Fibrosis Airways**)

Dear Dr. Isaac Martin:

Please make the following edits (P and L numbers correspond to the marked manuscript):

P 3 L 20 Better as "during biofilm formation"

P.5 L 18 Consider: Screening strains for biofilm disruption by PslGh and PelAh

P.5 L. 19 Rephrase "Psl and Pel dependence"

P. 14 L 14-19 Consider: Two representative clinical strains were selected based on their susceptibility to GH disruption - PA342 (preferentially disrupted by PslGh) and PA380 (preferentially disrupted by PelAh) - to assess whether GH activity relates to the targeted exopolysaccharide and whether higher enzyme concentrations (0.5 mg/mL versus 0.1 mg/mL) enhance the effects observed in the initial single-dose screen.

P. 22 L. 13-15, the sentence needs editing, as "PslGh-mediated disruption predominating" is not clear.

Your manuscript has been accepted, and I am forwarding it to the ASM production staff for publication. Your paper will first be checked to make sure all elements meet the technical requirements. ASM staff will contact you if anything needs to be revised before copyediting and production can begin. Otherwise, you will be notified when your proofs are ready to be viewed.

Sincerely,
Marianna Patrauchan
Editor
Microbiology Spectrum